# A symmetry-matching approach to blind-spot reduction in sparse autoencoders

## Abstract

Language models can treat semantically distinct inputs as interchangeable at the representation level, creating blind spots that no standard sparse autoencoder (SAE) objective is designed to detect. In safety-critical settings such as clinical dosage extraction, legal clause interpretation, or financial amount verification, such failures can propagate silently into downstream decisions. We study this problem as one of feature-basis orientation. Building on a symmetry-matching view from algebraic error-detection theory, we add a differentiable $V_{\mathrm{Gini}}$ regularisation term to SAE training that penalises uneven perturbation response within a specified family. Across GPT-2, Gemma 2, and Qwen 2.5, a frozen multi-seed evaluation shows that $V$-regularisation raises both relative and absolute lower-tail responses while reducing response-profile imbalance, with no material general-text reconstruction degradation. JumpReLU and minimum description length (MDL) objectives remain close to the Standard SAE profile under the same protocol. Family-geometry controls show that the effect is not explained by template duplication or a single dominant perturbation family, and held-out probes plus null-calibrated separation tests evaluate how far the representation-level change carries to downstream discriminability. A zero-shot audit on out-of-distribution clinical radiology minimal pairs finds the weak-tail lift in all evaluated model-by-family cells, moving independently of the trained statistic, and a held-out family audit finds it on family kinds never used in training. Together, the results show that perturbation-aware SAE training can reorient feature bases toward safety-relevant input changes while leaving downstream validation as a separate empirical step.

## 1 Introduction

Consider a clinical pipeline that must preserve the distinction between "the patient received 3 doses" and "the patient received 8 doses". In the learned feature space of a sparse autoencoder, such a numeric change can land in the weak lower tail of the perturbation-response profile, so the code moves very little for exactly the pairs where the change matters. For the dose value itself this weakness stays hidden, because the digit sits on the surface of the text and any downstream readout can recover it directly. The same weak-tail geometry has no such safety net for distinctions that no single surface token carries, such as negation, severity, or laterality, and there it can translate into degraded downstream access. This is not primarily a failure of knowledge. It is a failure of orientation in the feature space. The relevant distinction may be present in the hidden state, but poorly arranged in the feature basis, so that the perturbation induces too little code-space separation to support reliable access.

The same algebraic phenomenon appears in a seemingly unrelated domain. A decimal checksum that uses weighted modular arithmetic cannot detect certain transpositions of adjacent digits, because the arithmetic makes those transpositions invisible. The Hungarian patient identifier (TAJ) and several US healthcare identifiers (NPI, DEA, NDC) are built on exactly such checksums and inherit this blind spot (Balogh, 2026). The fix in both cases is not more redundancy but better orientation of the existing encoding.

The need for systematic perturbation-awareness testing in clinical AI is increasingly recognised. Moradi et al. (2021) demonstrated that state-of-the-art clinical NLP models degrade under minor input perturba-

tions that simulate real-world noise in clinical text, and the SemEval NLI4CT shared task (Jullien et al., 2024) attracted over 100 participating teams to evaluate biomedical inference robustness under controlled perturbations. The major AI developers acknowledge the problem. OpenAI's safety evaluation framework explicitly aims to "protect against blind spots" through third-party assessments (OpenAI, 2025), and Anthropic's interpretability programme has identified that not all sparse autoencoder features correspond to actionable representations, calling for improved methods to ensure that feature decompositions capture safety-relevant distinctions (Templeton et al., 2024; Olah et al., 2020). A joint Anthropic–OpenAI alignment evaluation exercise in 2025 (Anthropic, 2025) tested for misalignment-related blind spots across both companies' flagship models, underscoring the industry consensus that representational blind spots are a first-order safety concern. The present paper provides a family-specific, differentiable metric for perturbation awareness and turns it into an SAE training signal.

This paper connects the clinical, algebraic, and interpretability domains and argues that reliable artificial intelligence requires two independent conditions. The first is *knowledge*, addressed by the scaling-laws literature (Kaplan et al., 2020; Hoffmann et al., 2022) through larger models and more data. The second is *perturbation awareness*, the ability to respond to input changes that matter for the task. We start from the symmetry-matching condition $\mathrm{Stab}(G, F) \cap E = \{e\}$ of the algebraic framework for error-detecting codes over finite alphabets (Balogh, 2026), and Section 2 gives a self-contained account of the condition and its continuous adaptation. In continuous SAE representations the response profile has more structure than a binary visible/invisible distinction, so we evaluate response magnitude, lower-tail behaviour, and the Gini imbalance of the finite response profile.

We add $V_{\mathrm{Gini}}$ as a differentiable regularisation term to the sparse autoencoder (SAE) training objective and compare the result against three alternative SAE objectives (standard $L_1$, JumpReLU (Rajamanoharan et al., 2024), and MDL (minimum description length) (Ayonrinde & Pearce, 2024)) on three language models of different scale and architecture. We then extend the evaluation from six perturbation families to sixteen, add family-geometry controls, and test whether representation-level changes transfer to held-out and null-calibrated probes.

**Contributions.** (i) We make representational blind spots measurable and localisable by defining a perturbation-response hierarchy for continuous SAE representations, connecting exact blind sets, low-response regions, lower-tail response magnitudes, and finite response-profile imbalance. This turns perturbation awareness into an optimisable target (§2). (ii) We introduce $V_{\mathrm{Gini}}$ regularisation as a differentiable perturbation-sensitivity conditioning term for SAE training (§3). (iii) We evaluate the method under a frozen multi-seed protocol and show that it raises lower-tail perturbation responses while reducing sensitivity imbalance (§5). (iv) We show that three standard SAE objectives ($L_1$, JumpReLU, and MDL) are blind to perturbation-response imbalance. All three leave the profile close to the unregularised Standard SAE, establishing that perturbation awareness is not captured by reconstruction, sparsity, or description-length pressure. We add family-geometry, held-out, and null-calibrated validation checks (§5–§6). (v) We use a unified extraction protocol to ensure that perturbation responses are compared at matched sequence positions across GPT-2, Gemma 2, and Qwen 2.5.

**Related work.** Sparse autoencoders are the dominant tool for decomposing language-model activations into interpretable features (Olah et al., 2020; Elhage et al., 2022; Bricken et al., 2023; Cunningham et al., 2024; Templeton et al., 2024; Gao et al., 2024), and a growing literature questions whether the recovered features are canonical units of analysis (Paulo & Belrose, 2025; Tang, 2025; Kissane et al., 2024). This work is orthogonal to that debate. Rather than asking whether the features are the "right" ones, we ask whether the basis they span is oriented so that task-relevant perturbations are uniformly visible. The adversarial robustness literature (Goodfellow et al., 2015; Madry et al., 2018) measures the *magnitude* of perturbation needed to change a model's output; the $V$ measure instead characterises the *direction* of perturbations a representation under-responds to, complementing the geometric/equivariance view of deep networks (Bronstein et al., 2021).

## 2 Blind sets, perturbation responses, and sensitivity imbalance

### 2.1 The symmetry-matching condition

Every encoding has symmetries, meaning input transformations that leave the encoded state unchanged. In a checksum these are undetected transcription errors; in a neural representation they are input changes that the representation does not separate. We recall the algebraic condition that makes this notion precise and then adapt it to a continuous encoder.

Following Balogh (2026), let $G$ be a group acting on the input space $X$, with the action written $g \cdot x$ for $g \in G$ and $x \in X$. The elements of $G$ are the input transformations available to the channel. Let $F \colon X \to Z$ be the encoding, here the sparse-autoencoder encoder that maps a hidden state to its code, and let $E \subseteq G$ be the family of task-relevant transformations, the changes that a downstream task requires the representation to track. The stabiliser of the encoding is the set of transformations under which the code does not move,

$$\mathrm{Stab}(G, F) = \{g \in G \mid F(g \cdot x) = F(x) \text{ on the evaluated domain}\}. \tag{1}$$

A transformation in $\mathrm{Stab}(G, F)$ is invisible to the encoding. The symmetry-matching condition requires that no task-relevant transformation be invisible,

$$\mathrm{Stab}(G, F) \cap E = \{e\}, \tag{2}$$

where $e$ is the identity transformation. When the intersection contains a non-identity element, that element is a task-relevant change the encoding cannot see, and it is an exact blind spot. In the algebraic source of Eq. (2) the group is finite and the stabiliser is a finite set of sequence transformations, so the condition is decidable by enumeration. The remainder of this section carries the same condition into the continuous setting, where $F$ is a trained encoder and exact invariance is the boundary case of a graded response.

The experiments in this paper do not observe all of $E$. They sample a finite subfamily $E_{\mathrm{obs}} \subseteq E$ as a set of perturbation pairs $(x_i, g_i \cdot x_i)$ with $g_i \in E$, indexed by $i \in \{1, \dots, K\}$. Every quantity below is defined on this finite sample, and the algebraic condition (2) is the limiting statement that the quantities approximate.

### 2.2 From exact invariance to a response profile

For each sampled pair $i$, with codes $z_i^{\mathrm{orig}} = F(x_i)$ and $z_i^{\mathrm{pert}} = F(g_i \cdot x_i)$, define the relative SAE-code response

$$D_i = \frac{\|z_i^{\mathrm{pert}} - z_i^{\mathrm{orig}}\|}{\|z_i^{\mathrm{orig}}\| + \varepsilon}. \tag{3}$$

The response $D_i$ measures how far the code moves under the applied transformation, normalised by the code norm so that the quantity is comparable across pairs. The exact blind set on the sampled family is the set of transformations that move the code not at all,

$$S_0(F, E) = \{i \mid D_i = 0\}. \tag{4}$$

The emptiness of $S_0$ is the sampled counterpart of Eq. (2), since $D_i = 0$ means $g_i \in \mathrm{Stab}(G, F)$.

Exact invariance is the wrong operational target for a continuous encoder, for the following reason.

**Proposition 1** (Exact blind sets vanish almost surely). *Let $F$ be continuous and let the perturbation pairs $(x_i, g_i \cdot x_i)$ be drawn from a distribution that is absolutely continuous with respect to Lebesgue measure on a neighbourhood of the evaluated domain, with $g_i \cdot x_i \neq x_i$ almost surely. Suppose $F$ is not locally constant on any set of positive measure. Thus $D_i > 0$ almost surely, and therefore $S_0(F, E) = \emptyset$ with probability one.*

*Proof.* Fix a pair and write $x = x_i$, $x' = g_i \cdot x_i$. The event $D_i = 0$ is the event $F(x') = F(x)$. Consider the set $Z = \{(u, v) \mid F(u) = F(v)\} \subseteq X \times X$. Since $F$ is continuous, $Z$ is closed. By hypothesis $F$ is not locally constant on any positive-measure set, so the diagonal-avoiding part $Z \setminus \{u = v\}$ has Lebesgue measure zero in $X \times X$. The pair $(x, x')$ avoids the diagonal almost surely because $g_i \cdot x_i \neq x_i$ almost surely, and it is drawn

from an absolutely continuous distribution, so it lands in the measure-zero set $Z \setminus \{u = v\}$ with probability zero. Hence $F(x') \neq F(x)$ almost surely, which gives $D_i > 0$ almost surely. Taking the union over the finite index set $i \in \{1, \ldots, K\}$ preserves probability one, so $S_0(F, E) = \emptyset$ with probability one. □

Proposition 1 says that the exact algebraic level is operationally empty for a continuous encoder. A trained SAE will almost never leave a perturbation response at exactly zero, so a test built on $S_0$ reports triviality regardless of whether the representation separates the task-relevant change. The informative question is not whether the response is exactly zero, but whether it is small relative to a chosen detection scale, and how the responses are distributed across the family.

Two graded quantities replace the exact condition. For a fixed response margin $\gamma$, the low-response margin set collects the pairs with response below the margin,

$$S_\gamma(F, E) = \{i \mid D_i < \gamma\}, \tag{5}$$

and the coverage statistic reports the mass above the margin,

$$C_\gamma(F, E) = \frac{1}{|E|} \sum_{i \in E} \mathbf{1}[D_i \geq \gamma]. \tag{6}$$

As $\gamma \to 0$ the set $S_\gamma$ contracts to $S_0$, so the margin version is a continuous relaxation of the exact blind set, with $\gamma$ setting the detection scale. For margin-free summaries we use the lower tail $L_{20}(D)$, the mean of the smallest quintile of responses, and the absolute code displacement lower tail $L_{20}(\|\Delta z\|)$. The lower-tail quantities report the magnitude of the weakest responses, which is the regime a blind spot inhabits.

### 2.3 The Gini imbalance of the response profile

The margin set and the lower tail measure the size of weak responses. The regulariser acts on a complementary property, the inequality of the response profile across the family. For a family of responses $(D_1, \ldots, D_K)$ with mean $\bar{D} = K^{-1} \sum_i D_i$, define the imbalance as the mean absolute pairwise difference of responses, normalised by twice the mean,

$$V_{\text{Gini}}(D) = \frac{1}{2K^2 \bar{D}} \sum_{i=1}^{K} \sum_{j=1}^{K} |D_i - D_j|. \tag{7}$$

This form states directly what the metric measures. The numerator sums the response gap over every pair of transformations in the family, so it is large when a few directions absorb most of the response and the rest are weak, and it is zero when every direction responds equally. The normalisation by $\bar{D}$ removes the response scale, so the metric reports the shape of the profile rather than its size.

Equation (7) is the Gini coefficient of the response profile, and it has an equivalent sorted form that is cheaper to evaluate. Writing $D_{(1)} \leq \cdots \leq D_{(K)}$ for the sorted responses,

$$V_{\text{Gini}}(D) = \frac{2 \sum_{i=1}^{K} i \, D_{(i)}}{K \sum_{i=1}^{K} D_{(i)}} - \frac{K+1}{K}. \tag{8}$$

The two forms are equal for every nonnegative profile. The sorted form is used in training because it costs one sort rather than $K^2$ pairwise differences, and its two terms are not a design choice. They follow from the pairwise numerator by counting how often each response appears. Once the responses are sorted, the ordered value $D_{(i)}$ is larger than the $i-1$ responses below it and smaller than the $K-i$ responses above it, so in the double sum of absolute differences it enters with the coefficient $(i-1) - (K-i) = 2i - (K+1)$. Summing over ranks,

$$\sum_{i=1}^{K} \sum_{j=1}^{K} |D_i - D_j| = \sum_{i=1}^{K} (2i - (K+1)) D_{(i)} = 2 \sum_{i=1}^{K} i \, D_{(i)} - (K+1) \sum_{i=1}^{K} D_{(i)},$$

Table 1: Hierarchy of perturbation-response quantities used in this paper.

| Object | Measures | Role |
|---|---|---|
| $S_0$ | Exact blind set | Algebraic limit case |
| $S_\gamma$ | Low-response margin set | Operational response threshold |
| $C_\gamma$ | Coverage above a fixed margin | Mass above a chosen threshold |
| $V_{\text{Gini}}$ | Response-profile imbalance | SAE training objective |
| $L_{20}(D)$ | Lower-tail relative response | Weak-response lifting |
| $L_{20}(\|\Delta z\|)$ | Lower-tail absolute code displacement | Scale-sensitive support |
| Held-out probes | Discriminability evidence | Downstream transfer check |
| Null-calibrated AUROC | Separation from nuisance controls | Calibrated representation check |

and dividing by $2K \sum_i D_{(i)}$ produces the two terms of Eq. (8). The first term is therefore the rank-weighted mean of the sorted responses, which grows when large responses sit at high ranks and small responses at low ranks, that is when the profile is unequal. The second term $\frac{K+1}{K}$ is the value the first term takes on a perfectly equal profile, so the subtraction sets the metric to zero at equality and lets it grow toward one as the profile concentrates. The rank multiplicity $2i - (K + 1)$ is the reason the metric takes a difference of two terms rather than a single sum. Appendix A records the boundary behaviour of the identity and the differentiable surrogate used for training.

Two properties fix the interpretation. First, $V_{\text{Gini}}(D) = 0$ if and only if all responses are equal, and it approaches its maximum when the response concentrates on a single transformation, so it is a measure of concentration, not of magnitude. Second, it is invariant to common rescaling of the profile,

$$V_{\text{Gini}}(cD_1, \ldots, cD_K) = V_{\text{Gini}}(D_1, \ldots, D_K) \quad \text{for all } c > 0, \tag{9}$$

which follows because the factor $c$ cancels between numerator and denominator in Eq. (7). A representation can therefore have a low $V_{\text{Gini}}$ and still respond weakly in absolute terms if every response is uniformly small. The scale-invariant statistic is the training target because it is the quantity a differentiable objective can shape, and the lower-tail quantities $L_{20}(D)$ and $L_{20}(\|\Delta z\|)$ carry the magnitude information that Eq. (9) discards. We report the two together throughout, and we do not read a low $V_{\text{Gini}}$ on its own as evidence of detectability. For calibrated separation checks, we report area under the receiver operating characteristic curve (AUROC). The hierarchy is summarised in Table 1.

## 3   $V$-regularised SAE training

The augmented training loss is

$$\mathcal{L} = \|h - \hat{h}\|^2 + \lambda_1 \|z\|_1 + \lambda_2 \bar{V}_{\text{Gini}}(z, E), \tag{10}$$

where $\bar{V}_{\text{Gini}}$ averages over sampled perturbation families. Perturbation hidden states are pre-cached (one forward pass through the frozen language model per perturbation pair) so that the $V$ term adds negligible training cost and does not introduce stochastic variation across steps (Appendix T). The base model's weights remain frozen; only the SAE feature basis is modified.

## 4   Experimental setup

**Models.**   GPT-2 small (Radford et al., 2019) (124M parameters, hidden dim 768, layer 12), Gemma 2 2B (Gemma Team, 2024) (2.6B, hidden dim 2,304, layer 13), and Qwen 2.5 3B (Qwen Team, 2024) (3B, hidden dim 2,048, layer 18). All SAEs use $d_{\text{sae}} = 4,096$ with ReLU activation (standard and $V$-regularised variants), JumpReLU activation (JumpReLU variant), or ReLU with an MDL penalty (MDL variant).

**Hidden-state extraction.**   For decoder-only models with different padding conventions, hidden states are extracted using a true-last-token rule computed from the attention mask. This ensures that perturbation

Table 2: Frozen multi-seed Standard versus $V$-regularised SAE comparison. Values are mean $\pm$ standard deviation across three seeds. Positive $\Delta L_{20}(D)$ and $\Delta L_{20}(\|\Delta z\|)$ indicate lower-tail response lifting; negative $\Delta V_{\text{Gini}}$ indicates reduced response-profile imbalance. The OWT reconstruction change is reported as NMSE difference.

| Model | $\Delta L_{20}(D)$ | $\Delta L_{20}(\|\Delta z\|)$ | $\Delta V_{\text{Gini}}$ | Positive families | $\Delta$OWT NMSE |
|---|---|---|---|---|---|
| GPT-2 | $+0.0562 \pm 0.0010$ | $+5.3584 \pm 0.1202$ | $-0.2919 \pm 0.0008$ | $1.0000 \pm 0.0000$ | $+0.0000 \pm 0.0000$ |
| Gemma 2 | $+0.1301 \pm 0.0042$ | $+10.3454 \pm 1.1553$ | $-0.2322 \pm 0.0001$ | $1.0000 \pm 0.0000$ | $-0.0004 \pm 0.0007$ |
| Qwen 2.5 | $+0.1316 \pm 0.0006$ | $+2.4650 \pm 0.1129$ | $-0.2432 \pm 0.0001$ | $1.0000 \pm 0.0000$ | $+0.0006 \pm 0.0011$ |

responses are compared at the final non-padding token position across GPT-2, Gemma 2, and Qwen 2.5. Reconstruction quality is measured on OpenWebText (OWT) hidden-state samples as the normalised mean squared error (NMSE) between the SAE input and its reconstruction. When raw reconstruction MSE is reported, it is labelled explicitly.

**Perturbation families.** We evaluate six task-relevant perturbation families. Some are adapted from classical edit-error taxonomies (Damerau, 1964; Pollock & Zamora, 1984), while others test semantic substitutions, negation, synonym changes, and numeric swaps. The six families are semantic substitution, typo, negation, synonym, number swap, and word order. Each contains 50 sentence pairs. For the generalisation study (§5.11) we add ten auto-generated medical-domain families (anatomical direction, body-part swap, causal reversal, condition negation, date/time change, drug-name swap, frequency change, gender swap, severity change, unit of measure) with 30 pairs each, giving sixteen families and 600 pairs total. A further pre-constructed pool of one hundred families with thirty fixed pairs each, of which eighty-four were never used in training, supports the held-out family audit (§5.4, Appendix E). Representative examples for all sixteen families are listed in Appendix U, and the complete datasets are provided as supplementary material (see Reproducibility and supplementary material).

**Training protocol.** All variants share $\lambda_1 = 10^{-3}$, learning rate $3 \times 10^{-4}$, batch size 64, 5,000 steps, and the Adam optimiser. The $V$-regularised variants use $\lambda_2 = 0.1$, sampling 3 families per step with 8 pairs each from the pre-cached perturbation hidden states. The sixteen-family "joint" runs use the script `train_joint.py` with all sixteen families in the $V$ loss at every step, $\lambda_2 = 0.2$, and 7,500 steps. All experiments run on a single Apple M-series GPU.

## 5 Results

### 5.1 Multi-seed evaluation across three model families

We first report a frozen multi-seed stability evaluation. It uses true-last-token perturbation caches for all three models. Table 2 reports seed-aggregated deltas for Standard versus $V$-regularised SAEs, and Figure 1 visualises how the weakest Standard responses move under $V$-regularisation. With this hierarchy in place, we interpret a lower Gini value together with the movement of the weak-response tail. Thus, we report the decrease in $V_{\text{Gini}}$ alongside positive $\Delta L_{20}(D)$ and positive $\Delta L_{20}(\|\Delta z\|)$. Across GPT-2, Gemma 2, and Qwen 2.5, $V$-regularisation raises lower-tail relative response and lower-tail absolute code displacement while reducing response-profile imbalance. The positive-family fraction is 1.0 in all three models, so the effect is not driven by a small subset of families.

Figure 1 gives the same result at the distributional level. $V$-regularisation shifts the empirical response distribution to the right, removing all or most perturbation pairs from the Standard bottom-20% response region. The remaining mass falls from 20% by construction to 0.0% on GPT-2, 5.5% on Gemma 2, and 0.0% on Qwen 2.5. Thus, the primary result is a three-model response profile effect, lower-tail lifting together with reduced within-family imbalance and no material OWT reconstruction degradation.

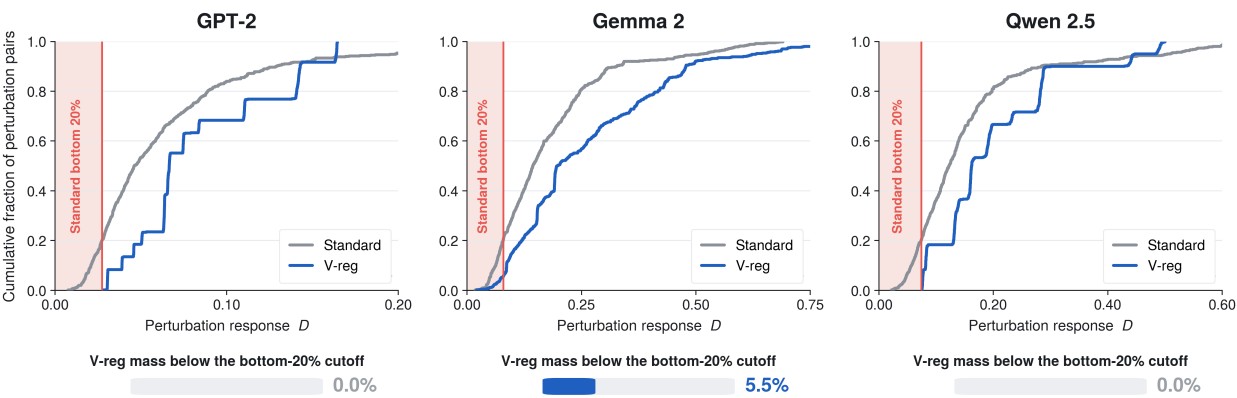

Figure 1: Empirical cumulative distribution functions (CDFs) of per-pair perturbation response $D$ over all sixteen perturbation families. For each model, the red line marks the 20th percentile of the Standard SAE response distribution, defining a model-specific low-response region. Lower CDF values to the left of the cutoff indicate fewer perturbation pairs remaining in the weak-response region under $V$-regularisation. The step structure of the V-reg curves reflects the resolution of each model's code space. GPT-2 small concentrates pairs on fewer discrete response levels, producing visible steps, while the higher-dimensional Gemma 2 spreads responses almost continuously.

Table 3: Lower-tail response and absolute code displacement on the sixteen-family perturbation set. Values are V-reg minus Standard for the corresponding checkpoint pair. Positive $\Delta L_{20}(D)$ and $\Delta L_{20}(\|\Delta z\|)$ indicate that the weak lower tail is lifted in relative and absolute SAE-code response. The gain-normalised lower tail records the input-complexity-normalised variant.

| Model | $\Delta L_{20}(D)$ | Positive families | $\Delta L_{20}(\|\Delta z\|)$ | $\Delta L_{20}(\text{decoded})$ | $\Delta L_{20}(g)$ |
|---|---|---|---|---|---|
| GPT-2 | +0.055845 | 16/16 | +5.4349 | +0.000368 | −0.159311 |
| Gemma 2 | +0.133151 | 16/16 | +11.0187 | +0.067886 | −0.151532 |
| Qwen 2.5 | +0.131023 | 16/16 | +2.4443 | +0.040904 | −0.107768 |

## 5.2 Lower-tail response and absolute code displacement

The absolute-response analysis in Table 3 reports the magnitude side of the hierarchy. It gives the lower tail of relative SAE-code response, the lower tail of absolute code displacement, decoded-response support, and input-normalised gain on the sixteen-family perturbation set.

Across all three models, the lower tail of relative response and the lower tail of absolute code displacement increase in all sixteen families. The decoded-response column is positive for Gemma 2 and Qwen 2.5 and close to zero for GPT-2, so the representation-level lift has different decoded expression across models. The input-normalised gain column separates this main effect from a stricter gain-normalised variant.

## 5.3 Family geometry and homogeneity controls

The sixteen-family result could be misleading if it were driven by duplicated templates, one dominant family, or unusually homogeneous perturbation geometry. We therefore built exact-skeleton template clusters for all 600 perturbation pairs and recomputed uncertainty with a fixed-family template-cluster bootstrap. The collection contains 424 template clusters; 250 of 600 pairs (41.7%) belong to repeated templates.

Table 4: Template-cluster and family-resampled uncertainty for the lower-tail effects. Fixed intervals resample template clusters within each of the sixteen families. Family-resampled intervals additionally resample families and are wider, but remain positive for the relative lower-tail effect.

| Model | Metric | Fixed-family interval | Family-resampled interval |
|-------|--------|----------------------|---------------------------|
| Gemma 2 | $\Delta L_{20}(D)$ | $+0.1332\,[+0.1256, +0.1361]$ | $[+0.0891, +0.1811]$ |
| GPT-2 | $\Delta L_{20}(D)$ | $+0.0558\,[+0.0527, +0.0567]$ | $[+0.0417, +0.0699]$ |
| Qwen 2.5 | $\Delta L_{20}(D)$ | $+0.1310\,[+0.1248, +0.1328]$ | $[+0.0868, +0.1792]$ |
| Gemma 2 | $\Delta L_{20}(\|\Delta z\|)$ | $+11.0187\,[+9.8028, +11.7425]$ | $[+6.2071, +16.3177]$ |
| GPT-2 | $\Delta L_{20}(\|\Delta z\|)$ | $+5.4349\,[+5.1541, +5.5791]$ | $[+4.0521, +6.9244]$ |
| Qwen 2.5 | $\Delta L_{20}(\|\Delta z\|)$ | $+2.4443\,[+2.1110, +2.7675]$ | $[+0.9824, +4.1303]$ |

The relative lower-tail effect remains positive under every leave-one-template-out deletion across 424/424 template clusters for all three models, and leave-one-family-out macro effects remain positive. The hidden perturbation geometry is well-defined under the true-last-token extraction rule for every model and family. GPT-2 shows a more low-dimensional perturbation geometry than Gemma 2 or Qwen 2.5, but this geometric difference does not remove the positive sparse-code lower-tail effect.

## 5.4 Extrinsic endpoints beyond the training objective

The objective optimises a scale-invariant imbalance statistic, so a reduction in $V_{\mathrm{Gini}}$ alone does not establish that perturbations become more accessible. This subsection therefore reports endpoints that are not the training target. Each is measured on frozen checkpoints or a controlled paired design, and each is either independent of the response function or measures its absolute magnitude rather than its shape.

A null-calibrated separation test measures code-space distance against meaning-preserving nuisance controls rather than a zero-distance null. The primary endpoint is the family-macro AUROC difference between $V$-regularised and Standard SAEs on absolute code distance (Table 5). Gemma 2 and Qwen 2.5 show positive separation under the fixed-family interval, whereas GPT-2 is uncertain in point direction. The endpoint is a separation task the objective never sees, so it is external to the framework that defines the training target.

Held-out probes test whether the representation-level change reaches a semantic distinction on templates unseen during training. On the held-out negation probe the distinction is fully present in the Qwen 2.5 hidden state at an AUROC of 1.00, the Standard SAE code loses part of it at 0.85, and the $V$-regularised code recovers it to 0.91. This is the accessibility pattern the weak-tail signature predicts, closed on an endpoint outside the objective, and it appears where the Standard code has room to improve. On held-out negation templates the paired lower-tail probability-margin point estimates also improve for all three models, rising from 0.0052 to 0.3627 on GPT-2, from 0.1590 to 0.3646 on Gemma 2, and from 0.0846 to 0.1440 on Qwen 2.5, with family-stratified bootstrap intervals that include zero, so the margin version is suggestive rather than uniform.

A real-clinical audit evaluates the frozen general checkpoints zero-shot on OpenI chest radiography minimal pairs in four clinical families, negation, laterality, severity, and anatomical direction. The lower-tail response $L_{20}(D)$ rises in all eight model-by-family cells, with report-clustered intervals excluding zero in each, at macro means of $+0.0023$ on GPT-2 and $+0.0037$ on Qwen 2.5. The audit covers GPT-2 and Qwen 2.5; Gemma 2 enters the external OpenI evaluation through the laterality probe below and through the held-out family audit of Appendix E, where every Gemma 2 family is individually interval-positive. On this data the lower-tail lift and the trained inequality statistic move independently. The largest Qwen lift, negation at $+0.0099$, accompanies a $V_{\mathrm{Gini}}$ change of only $-0.003$, while the largest GPT-2 inequality drop, anatomical direction at $-0.056$, accompanies the smallest lift at $+0.0008$. Because $V_{\mathrm{Gini}}$ is scale-invariant by Eq. (9), it cannot by construction raise the absolute tail magnitude, and the data confirm that the magnitude gain is separate from the shape change the objective targets. Appendix B gives the per-family table and a qualitative case study on concrete minimal pairs.

Table 5: Null-calibrated separation. Family-macro AUROC difference (V-reg minus Standard) for absolute SAE-code distance against meaning-preserving nuisance controls. Fixed-family intervals resample template clusters within each family; family-resampled intervals also resample families. Intervals are 95% confidence intervals (CIs). Gemma 2 and Qwen 2.5 are positive under the fixed-family interval; GPT-2 is uncertain in point direction.

| Model | $\Delta$AUROC (abs. code) | Fixed-family 95% CI | Family-resampled 95% CI |
|---|---|---|---|
| Gemma 2 | +0.037 | [+0.013, +0.063] | [−0.014, +0.087] |
| Qwen 2.5 | +0.048 | [+0.024, +0.077] | [+0.007, +0.091] |
| GPT-2 | −0.010 | [−0.049, +0.024] | [−0.078, +0.064] |

Two controlled designs isolate the mechanism. A dedicated paired experiment trains Standard and $V$-regularised SAEs from identical initialisation and an identical reconstruction minibatch stream, differing only in the $V$ term. The regulariser drives the trained family's response inequality to near zero, from 0.233 to 0.0009 on GPT-2 and from 0.113 to 0.0008 on Qwen 2.5, with no measurable reconstruction cost on held OpenWebText tokens, where explained variance is unchanged on GPT-2 and rises from 0.995 to 0.996 on Qwen 2.5 (Appendix C). A controlled synthetic experiment then places the task in a non-saturated regime by construction. There the $V$-regularised code yields a modest held-out probe gain of +0.016 in AUROC together with a clear lower-tail displacement gain of +0.51, and the probe gain shrinks as the Standard code approaches ceiling, matching the headroom pattern of the real models (Appendix D).

External OpenI laterality probes give a stricter boundary check. On left/right minimal pairs, the lower-tail probability-margin deltas are small, at +0.0022 for GPT-2, −0.0747 for Gemma 2, and +0.0141 for Qwen 2.5. The intervals cross zero for GPT-2 and Qwen, while the Gemma 2 delta is negative with a report-clustered interval of [−0.163, −0.012] that excludes zero, a downstream cost on this saturated external task. A targeted certificate-gated repair on the same task moves the intended laterality direction without a measurable downstream gain, consistent with this saturated boundary (Appendix I). Finally, a held-out family audit evaluates the same frozen checkpoints on seventy-two family kinds never used in training, drawn from a pre-constructed pool and verifiable as held out from the released training materials, and finds the lower-tail lift at the same order of magnitude, including on eighteen family kinds with no mechanical overlap with training (Appendix E).

The checks give a consistent reading. $V$-regularisation improves the geometry of weak response directions on endpoints outside the objective, and transfer to downstream probability margins depends on the task and model, appearing where the Standard code has headroom.

## 5.5 Response-profile imbalance under alternative SAE objectives

We trained five SAE variants on each of the three models and evaluated the $V_{\mathrm{Gini}}$ of each on the six perturbation families. The five variants are the standard SAE ($L_1$ sparsity), JumpReLU SAE (learnable threshold with an $L_0$ proxy, Rajamanoharan et al., 2024), MDL SAE (description-length penalty, Ayonrinde & Pearce, 2024), $V$-regularised SAE ($V_{\mathrm{Gini}}$ penalty on raw response), and $V$-regularised SAE with input normalisation ($V_{\mathrm{Gini}}$ penalty on complexity-controlled gain).

Appendix K reports the single-run numerical detail. On all three models, the standard, JumpReLU, and MDL SAEs produce similar $V$ profiles; the differences among these three methods are consistently below 0.02 across all families and models. None of them targets perturbation awareness, and they leave response-profile imbalance largely unchanged. In that descriptive comparison, the $V$-regularised SAE produces substantially lower response-profile imbalance than the non-$V$ objectives, so reconstruction, sparsity, and description-length objectives do not by construction condition the perturbation-response profile.

A frozen multi-seed baseline extension confirms this separation beyond a single run. Using the same true-last-token caches and seeds as §5.1, we trained JumpReLU and MDL across three seeds and compared them with the Standard and $V$-regularised checkpoints under the primary metric hierarchy (Table 6). JumpReLU

Multi-seed objective comparison across model families

Figure 2: Frozen multi-seed baseline extension across Standard, JumpReLU, MDL, and $V$-regularised SAEs. The non-perturbation-aware objectives stay close to Standard on lower-tail response and response-profile imbalance, whereas $V$-regularisation separates on both axes across GPT-2, Gemma 2, and Qwen 2.5.

Table 6: Frozen multi-seed baseline extension. Deltas are each objective minus the Standard SAE, mean $\pm$ standard deviation across three seeds, under the same protocol and caches as Table 2. JumpReLU and MDL stay near Standard, while $V$-regularisation separates on every model.

| Model | Objective | $\Delta L_{20}(D)$ | $\Delta L_{20}(\|\Delta z\|)$ | $\Delta V_{\text{Gini}}$ | Positive families |
|---|---|---|---|---|---|
| | JumpReLU | $-0.0001 \pm 0.0002$ | $+0.0498 \pm 0.0026$ | $-0.0009 \pm 0.0002$ | $0.354 \pm 0.308$ |
| GPT-2 | MDL | $-0.0003 \pm 0.0002$ | $+0.0315 \pm 0.0040$ | $+0.0001 \pm 0.0002$ | $0.063 \pm 0.108$ |
| | V-reg | $+0.0562 \pm 0.0010$ | $+5.3584 \pm 0.1202$ | $-0.2919 \pm 0.0008$ | $1.000 \pm 0.000$ |
| | JumpReLU | $-0.0005 \pm 0.0001$ | $+0.1767 \pm 0.0063$ | $-0.0001 \pm 0.0001$ | $0.042 \pm 0.036$ |
| Gemma 2 | MDL | $-0.0006 \pm 0.0001$ | $+0.1084 \pm 0.0011$ | $-0.0001 \pm 0.0000$ | $0.042 \pm 0.072$ |
| | V-reg | $+0.1301 \pm 0.0042$ | $+10.3454 \pm 1.1553$ | $-0.2322 \pm 0.0001$ | $1.000 \pm 0.000$ |
| | JumpReLU | $-0.0005 \pm 0.0001$ | $+0.1539 \pm 0.0072$ | $+0.0001 \pm 0.0001$ | $0.063 \pm 0.063$ |
| Qwen 2.5 | MDL | $-0.0007 \pm 0.0000$ | $+0.0926 \pm 0.0157$ | $+0.0003 \pm 0.0002$ | $0.042 \pm 0.036$ |
| | V-reg | $+0.1316 \pm 0.0006$ | $+2.4650 \pm 0.1129$ | $-0.2432 \pm 0.0001$ | $1.000 \pm 0.000$ |

and MDL track the Standard objective on all three models, moving $L_{20}(D)$ and $V_{\text{Gini}}$ by less than 0.001, while $V$-regularisation raises the lower-tail response and reduces imbalance by a far larger margin and hits a positive-family fraction of 1.0. None of the non-perturbation-aware baselines reproduces the joint pattern of lower-tail lifting and imbalance reduction. Figure 2 visualises this separation in the two quantities most directly tied to the central claim.

## 5.6 Consistency across three architectures

The three models differ in parameter count (124M to 3B), training data (English-only to multilingual), and architecture family (OpenAI, Google, Alibaba). The frozen multi-seed effect of §5.1 holds for all three, so the response-profile conditioning is not tied to a single scale, corpus, or architecture family. Within the single-run view, Qwen 2.5 reaches the lowest post-regularisation imbalance, which motivates the scale analysis next.

Table 7: Joint $V$-regularised SAE ($\lambda_2 = 0.1$, all six families simultaneously, 5,000 steps) on three models. Raw reconstruction MSE in parentheses. Values are descriptive single-run point estimates; entries printed as 0.000 are rounded to three decimals (Appendix P), and multi-seed variability for the primary lower-tail endpoints is reported in Table 2.

| Family | GPT-2 | Gemma 2 2B | Qwen 2.5 3B |
|---|---|---|---|
| Negation | 0.000 | 0.000 | 0.000 |
| Typo | 0.017 | 0.000 | 0.000 |
| Synonym | 0.019 | 0.000 | 0.002 |
| Number swap | 0.036 | 0.000 | 0.005 |
| Semantic sub. | 0.048 | 0.000 | 0.001 |
| Word order | 0.046 | 0.002 | 0.000 |
| Mean | 0.028 | 0.0003 | 0.001 |
| MSE | (0.005) | (0.005) | (0.010) |

## 5.7 Scale-dependence of the numeric blind spot

The number swap family is the most safety-relevant of the six, for the clinical reason set out in §1. On GPT-2 (124M), the $V$-regularised SAE reduces the number swap imbalance from 0.264 to 0.034, on Gemma 2 (2B) from 0.287 to 0.004, and on Qwen 2.5 (3B) the post-regularisation value is 0.005. On Gemma 2, the mean number-swap sensitivity is essentially unchanged (0.073 to 0.075) while the lower tail roughly doubles (0.035 to 0.074), so the reduction reflects lifted weak directions rather than uniform suppression. These descriptive values are consistent with the broader view that scaling can affect the information available to the SAE, while the SAE objective affects how that information is arranged in the feature basis.

## 5.8 A single orientation for all families

We trained single-family SAEs (each regularised on one family only) and evaluated them on all six families. Every single-family SAE reduces imbalance on its target but leaves the other families largely unchanged. A joint SAE, regularised on all six families simultaneously, reaches a low-imbalance regime across the six-family set (Table 7; the Qwen column uses the method-comparison V-reg result, which regularises all six families at every step). The minimum number of parallel feature bases needed for simultaneous $V \approx 0$ is $m^* = 1$, meaning the six families do not conflict algebraically and a single well-oriented basis suffices.

In the check-digit framework (Balogh, 2026), this is analogous to a permutation parameter that simultaneously detects substitutions, transpositions, and twin errors with a single fold of the dihedral group $D_n^m$. The value $m^* = 1$ means that the six perturbation families tested here can coexist in one feature basis without algebraic conflict. When new families are added, the same protocol determines whether the enlarged set still admits $m^* = 1$ or requires $m^* > 1$; in the latter case the multi-factor architecture (Appendix J) provides $m$ parallel encoders at constant total capacity. The stabiliser monotonicity theorem of Balogh (2026) proves that adding factors can only shrink the stabiliser in the algebraic setting, so $m^*$ quantifies family compatibility for the chosen perturbation set.

## 5.9 Sensitivity increase, not suppression

A natural concern is that low $V$ might be achieved by suppressing all sensitivity (making every pair equally invisible). The per-family control in Appendix L shows the opposite. On all three models, the mean sensitivity *increases* for most families under $V$-regularisation. On GPT-2, negation sensitivity rises by 58% and word order by 34%. On Gemma 2, typo sensitivity more than triples (+219%) and number swap more than doubles (+154%). On Qwen 2.5, every family shows increased mean sensitivity. The feature basis becomes more uniformly alert, not blind.

Table 8: Secondary low-imbalance summary for sixteen-family joint $V$-regularised SAE runs. "Std SAE" and "V-reg" are the mean $V_{\text{Gini}}$ over all sixteen families; "$V{<}0.05$" and "$V{<}0.005$" are descriptive low-imbalance counts. This summary covers the two models for which the sixteen-family joint run was conducted; Gemma 2 is evaluated under the frozen multi-seed protocol in Table 2. Source files `results/joint_eval_{gpt2,qwen-2.5-3b}.json`.

| Model | Layer | Std SAE | V-reg | Reduction | $V{<}0.05$ | $V{<}0.005$ |
|---|---|---|---|---|---|---|
| GPT-2 small | 12 | 0.314 | **0.002** | $-99.3\%$ | 16/16 | 15/16 |
| Qwen 2.5 3B | 18 | 0.239 | **0.000** | $-100\%$ | 16/16 | 16/16 |

## 5.10 Input-normalised gain controls for perturbation complexity

The six perturbation families differ in complexity (a typo changes one character; a semantic substitution changes a content word with variable embedding distance). To verify that the $V$ reduction is not an artefact of within-family complexity homogeneity, we also compute the input-normalised gain metric (§2), which divides the feature-space sensitivity by the hidden-state perturbation size. The $V$-regularised SAE with gain normalisation (V-reg$_{\text{n}}$) achieves $V_{\text{gain}}$ values that round to 0.000 on all six families on all three models (Appendix M), indicating very low gain imbalance after input-complexity normalisation. This supports the interpretation that the effect is not merely complexity homogenisation.

## 5.11 Generalisation to sixteen perturbation families

To test whether the result is specific to the six families used above, we extended the joint $V$-regularised training to sixteen families, the six standard families plus ten auto-generated medical-domain families (§4). All sixteen families enter the $V$ loss at every step (script `train_joint.py`, $\lambda_2 = 0.2$, 7,500 steps; evaluation uses 50 pairs for each of the six original families and 30 for each of the ten medical families).

Table 8 reports a secondary low-imbalance summary for the sixteen-family joint runs. On GPT-2 and Qwen, the $V$-regularisation drives the mean $V$ across all sixteen families to a low-imbalance regime. GPT-2 falls from 0.314 to 0.002 ($-99.3\%$, with 16/16 families below 0.05 and 15/16 below 0.005), and Qwen falls from 0.239 to 0.000 ($-100\%$, with 16/16 families below 0.005).

The extension is informative because the *ten additional medical-domain families*, which were never used to design the metric or the training protocol, are reduced at least as thoroughly as the six original families. On GPT-2 the ten medical families fall from a mean $V$ of 0.322 to 0.000, marginally better than the six original families ($0.302 \rightarrow 0.006$), and on Qwen both groups reach 0.000 (from 0.251 and 0.218 respectively). Clinically salient families such as `severity_change` (GPT-2 $0.428 \rightarrow 0.000$), `frequency_change` ($0.377 \rightarrow 0.000$), `drug_name_swap` ($0.266 \rightarrow 0.000$), and `unit_of_measure` ($0.296 \rightarrow 0.000$) reach low imbalance under the joint run. Thus, the orientation generalises across perturbation type rather than only across the families used to derive it, which is the property a single well-oriented basis would be expected to show. The full per-family values are reported in Appendix R.

# 6 Discussion

The central finding is that an SAE feature basis can be reoriented at fixed capacity to redistribute perturbation sensitivity. The five variants tested here share the same width and training data and differ only in the training objective. Objectives that optimise reconstruction, sparsity, or description length leave the perturbation-response profile close to the Standard SAE. $V$-regularisation changes that profile directly.

This is the same orientation problem that motivates the checksum analogy in the introduction. A representation can contain enough information for a distinction to matter and still arrange its feature basis so that some perturbation directions are weakly expressed. The multi-seed result shows that the weak lower tail can be lifted across GPT-2, Gemma 2, and Qwen 2.5, while the sixteen-family runs show that the same

orientation can extend to additional medical-domain perturbation families included in the sixteen-family joint run. The family-geometry controls rule out the simplest explanations based on repeated templates or one dominant family.

For AI development practice, the method adds a concrete evaluation and training axis to the usual reconstruction and sparsity criteria. A team building a clinical-facing language model could specify safety-relevant perturbation families such as dosage changes, negation of contraindications, or unit-of-measure substitutions, measure the $V$ profile of the feature decomposition, and apply $V$-regularisation to families with high imbalance. The intervention modifies only the SAE training loss, not the base model or the deployment stack.

The practical stakes are broad. The WHO projects a global shortage of 11 million health workers by 2030 (World Health Organization, 2025), and clinical workflows may increasingly depend on AI-assisted pipelines with limited human oversight. In such settings, a corrupted or adversarially altered input, for example "80 mg" instead of "8 mg", can propagate if the internal representation treats the two values too similarly. A $V$ profile gives developers a way to locate the perturbation families and model layers where this weakness is concentrated. It can guide targeted SAE regularisation, identify perturbation families for additional evaluation, and inform perturbation-sensitive SAE retraining for the affected family. In that role, the metric contributes to safer AI deployment by making sensitivity gaps measurable and actionable, while downstream task validation remains the final test of application behaviour.

The held-out, calibrated, and external probes place this representation result in context. They show that improving the sparse-code response profile can align with downstream separation, especially for Gemma 2 and Qwen 2.5 under the null-calibrated code-distance test, but they also show that probability-margin transfer is task-specific. This is why the metric hierarchy in §2 is useful. $V_{\text{Gini}}$ controls the shape of the response profile, lower-tail and absolute-response metrics check that weak directions have been lifted, and downstream probes measure task accessibility. This interpretation rests on a precondition. $V$-regularisation reorients the feature basis but cannot create information that the hidden state does not encode. When the relevant distinction is not represented at the chosen layer, no change to the SAE objective can recover it, and the limitation lies in the hidden state rather than in the regulariser.

**Sensitivity to the choice of perturbation family.** The conclusions depend on the perturbation families the practitioner specifies, and this dependence has two sides. On the training side, the sixteen-family study and the family-geometry controls show that the low-imbalance regime is not driven by a single dominant family or by template duplication. On the evaluation side, the held-out family audit (Appendix E) evaluates the frozen checkpoints on a pre-constructed pool of one hundred families, of which eighty-four were never used in training, and finds the lower-tail lift at the same order of magnitude on unseen instances of seen kinds and on eighteen structurally novel kinds. The lift is therefore a property of the learned basis rather than of the specific family list, while the downstream ceiling cases show that the choice of family still determines whether the lift has room to translate into separability. Specifying families remains a modelling decision, and the framework treats it as an explicit input rather than a hidden assumption.

**Scope and negative results.** The effect is scoped to endpoints and models with headroom in the Standard code. On a numeric dose-value probe the surface token is linearly decodable and the Standard code is at ceiling, so $V$-regularisation is slightly worse there, with a $\Delta$AUROC between $-0.04$ and $-0.16$ across the three models (Appendix F). At the representation level the numeric family shows the same weak lower tail as the other families and the objective lifts it, yet at the readout level the dose value remains at ceiling because the digit is surface-decodable, which is why a value probe is not the informative downstream test and why the safety concern applies where this surface redundancy is absent. On real OpenI negation under report-grouped cross-validation the Standard code is only marginally sub-ceiling and the $V$-regularised code does not reliably improve it, with a $\Delta$AUROC interval of $[-0.002, +0.008]$ that includes zero, a null result. Where the Standard code already separates a family at ceiling, a focused single-family objective can reduce held-out linear separability rather than improve it, from an AUROC of 1.00 to 0.81 on the dedicated Qwen negation run. On Qwen the real-clinical lower-tail lift is accompanied by a higher reconstruction error in the perturbation-pair domain, with per-family normalised mean squared error rising from about 0.006–0.008 to about 0.040–0.044, while general OpenWebText reconstruction is unchanged. These cases share one

explanation. With frozen general checkpoints, tasks that are saturated in the Standard code or carried by a trivially decodable surface token leave no accessibility gap to close. The gains we report are therefore specific to endpoints with demonstrated headroom, and $V_{\text{Gini}}$ is not a standalone certificate of downstream separability. Two preflight checks give the same reading before any focused training. A numeric orientation stress test finds no clean non-saturated regime on Qwen 2.5 (Appendix G), and a severity headroom preflight finds the severity family already saturated in the Standard code on all three models (Appendix H). These boundary cases delimit where the method helps, and they leave the broader reading unchanged, which is that a representational blind spot can be defined, measured, and conditioned as a computable quantity, with the gains concentrating where the representation has room to improve.

Beyond this empirical scope, the framework connects naturally to cognitive science. Human perceptual blind spots, including change blindness and inattentional blindness (Simons & Chabris, 1999), can be interpreted through the same algebraic lens. Input changes that fall in the stabiliser of the current attentional state remain undetected. Looking forward, SAE training should be viewed as a multi-objective problem over reconstruction, sparsity, perturbation awareness, and compressibility. More broadly, turning a representational blind spot from an informal intuition into a measurable, differentiable criterion is a precondition for automation. An algorithm cannot search for blind spots that are not first defined as a computable quantity. By specifying perturbation response as a differentiable objective, this work makes perturbation awareness an optimisable target and, in principle, a target that future methods could discover and audit without a pre-specified family set, rather than one supplied by hand as it is here. Mapping the surface $V(\lambda_2, d_{\text{sae}}, \text{scale})$ and determining the orientation cost $m^*$ for broader perturbation taxonomies are natural next steps.

## 7  Conclusion

Representational blind spots in language models can be reduced by conditioning the SAE feature basis, without retraining the base model. A single differentiable regularisation term, optimising the Gini coefficient of perturbation responses, reduces response-profile imbalance and lifts weak perturbation directions at fixed SAE width. Across GPT-2, Gemma 2, and Qwen 2.5, the frozen multi-seed protocol shows consistent lower-tail lifting and imbalance reduction. On GPT-2 and Qwen 2.5, the sixteen-family joint runs drive the mean $V$ across additional medical-domain perturbation families to a low-imbalance regime. The result is not that one metric settles downstream safety, but that perturbation awareness is an actionable training target for feature decompositions.

### Broader Impact Statement

This work targets a safety-relevant failure mode of language models used in clinical and other high-stakes settings. The relevant failure is weak or uneven representational separation for input changes that matter for the task, such as numeric dosages, negations, or unit-of-measure substitutions. The $V$ measure and its regularisation are intended to make sensitivity imbalance measurable and reducible, and we expect the primary impact to be positive. The method can help developers inspect and reduce response-profile vulnerabilities before deployment. The $V$ profile should complement downstream clinical validation and human oversight. In particular, $V_{\text{Gini}}$ measures within-family response uniformity rather than absolute task detectability. A system that is uniformly insensitive across a safety-relevant family could still have low $V_{\text{Gini}}$, so any certification use would require absolute-sensitivity thresholds or downstream validation in addition to the Gini profile. The perturbation families used here are synthetic and English-language; deployment in other languages or clinical sub-domains requires deriving family-specific perturbation sets for that setting.

## Reproducibility and supplementary material

This paper is self-contained. The main text states all settings needed to reproduce the main results, and Appendices A–U give the full derivations, family-geometry controls, reporting conventions, protocol provenance, per-family and per-axis tables, and example sentence pairs for all sixteen perturbation families.

As supplementary material, we provide the datasets and splits used for the perturbation probes, the machine-readable result files behind the reported tables and figures, the protocol manifests for the frozen multi-seed

experiments, and the code used for cache building, training, evaluation, template-cluster resampling, and null-calibrated analyses. The supplementary archive also includes the OpenI-derived minimal-pair splits used for the external checks and a manifest mapping result files to the tables and figures in the paper. The pipeline uses PyTorch and the HuggingFace `transformers` library and reproduces on a single consumer GPU. Activation caches, feature caches, and SAE checkpoints are regenerated by the scripts on first run and are therefore not bundled.

## AI Declaration

During the preparation of this work Anthropic Claude was used to assist in refining the linguistic and mathematical prose and in drafting the companion training and evaluation scripts. The conceptual framework, the results, and the experimental programme are the authors' own. All content was reviewed and edited by the authors, who take full responsibility for the published article.

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

## A  Differentiability of $V_{\mathrm{Gini}}$

Section 2.3 derives the sorted form Eq. (8) from the pairwise form Eq. (7) through the rank multiplicity $2i - (K+1)$. We record here the boundary behaviour the derivation assumes. The identity holds for every nonnegative profile with $\bar{D} > 0$. When $\bar{D} = 0$ every response is zero, the profile is exactly equal, and both forms are assigned the value $V_{\mathrm{Gini}} = 0$ by convention, the continuous extension of Eq. (7) as $\bar{D} \to 0$ along nonnegative profiles. The metric is bounded in $[0,1)$, the value 0 is attained on the equal profile, and the supremum is approached as the response mass concentrates on a single transformation. Because the responses are nonnegative by construction, no sign correction is needed, and the ordering used in the rank argument is well defined up to ties, which do not change the value of either sum.

The Gini coefficient (Eq. 8) involves a sort, which is not differentiable in the classical sense because infinitesimal perturbations do not change the ordering. However, $V_{\text{Gini}}$ is a piecewise-linear function of the unsorted responses $D_1, \ldots, D_K$, and its subgradient is well-defined almost everywhere (undefined only when two values are exactly equal, a measure-zero event). In our implementation we use `torch.sort`, which returns both the sorted values and the permutation; its backward pass propagates gradients to the unsorted inputs via the inverse permutation, equivalent to the straight-through estimator for discrete permutations and standard in differentiable sorting (Blondel et al., 2020).

The gradient of $V_{\text{Gini}}$ with respect to the unsorted response $D_j$ is

$$\frac{\partial V_{\text{Gini}}}{\partial D_j} = \frac{2\,\pi(j)}{K \cdot S} - \frac{2\sum_{i=1}^{K} i\,D_{(i)}}{K \cdot S^2}, \tag{11}$$

where $\pi(j)$ is the rank of $D_j$ and $S = \sum_i D_i$. It is well-defined whenever $S > 0$ and no two values are exactly equal, both of which hold in practice due to the $\varepsilon$ in Eq. (3) and the continuous nature of the hidden-state activations. The gradient propagates through the encoder by the chain rule, with the ReLU non-differentiability at zero handled by the standard subgradient convention.

## B  Real-clinical lower-tail audit and qualitative case study

This appendix supports the real-clinical audit of §5.4 with the per-family table and a qualitative view of what a weak response looks like on concrete minimal pairs. The evaluation uses the frozen sixteen-family general checkpoints, zero-shot, on OpenI chest radiography reports.

**What a blind spot looks like.**   A clinical blind spot is a minimal pair that differs in a task-relevant way while the code barely moves. Consider a negation pair and a laterality pair drawn from the OpenI reports,

> *Affirmed.* "There is a small pleural effusion."
> *Negated.* "There is no pleural effusion."
>
> *Left.* "Opacity in the left lower lobe."
> *Right.* "Opacity in the right lower lobe."

Under the Standard SAE code the weakest responses in these families sit low. The lower-tail response $L_{20}(D)$ is 0.030 for negation and 0.0034 for laterality on Qwen 2.5, and 0.016 and 0.0024 on GPT-2. These weak-tail values are the code-space signature of a blind spot, meaning that for the hardest pairs in the family the code moves little when the clinical meaning flips.

**How the regulariser changes it.**   On the same frozen checkpoints the $V$-regularised code raises the weak tail in every family and on both models (Table 9). The lower tail rises in all eight model-by-family cells, and the report-clustered interval excludes zero in each. The largest lift is on negation for Qwen 2.5, where $L_{20}(D)$ moves from 0.030 to 0.040.

**The lift is not a by-product of the trained statistic.**   The lower-tail response and the trained inequality statistic move independently on this data. On Qwen 2.5 the largest lower-tail lift, negation at +0.0099, accompanies the smallest change in $V_{\text{Gini}}$, from 0.427 to 0.424. On GPT-2 the largest drop in $V_{\text{Gini}}$, anatomical direction from 0.644 to 0.588, accompanies the smallest lower-tail lift, +0.0008. The rank order of the lower-tail lift does not follow the rank order of the inequality change, so the magnitude gain is a separate effect from the shape change the objective targets, as Eq. (9) requires.

**Scope.**   The analysis stays at the level of the response profile, since no feature-activation study is included, so it supports no claim about monosemanticity of individual units or about a universal downstream gain. On Qwen 2.5 the lower-tail lift in this domain is accompanied by a higher perturbation-pair reconstruction error, reported in the scope paragraph of §6. The case study establishes the narrower and verifiable point, that pairs receiving little code displacement under the Standard SAE receive larger and more evenly distributed responses under $V$-regularisation on real clinical text.

Table 9: Real-clinical lower-tail response on OpenI minimal pairs, frozen general checkpoints, zero-shot. $L_{20}(D)$ is the mean relative code displacement over the weakest quintile of pairs. Intervals are report-clustered 95% intervals of $\Delta L_{20}$.

| Model | Family | $L_{20}$ Standard | $L_{20}$ V-reg | $\Delta L_{20}$ [95% CI] |
|---|---|---|---|---|
| GPT-2 | negation | 0.0157 | 0.0200 | +0.0042 [+0.0030, +0.0053] |
| GPT-2 | laterality | 0.0024 | 0.0034 | +0.0009 [+0.0008, +0.0011] |
| GPT-2 | severity | 0.0044 | 0.0077 | +0.0032 [+0.0027, +0.0038] |
| GPT-2 | anat. dir. | 0.0017 | 0.0025 | +0.0008 [+0.0007, +0.0010] |
| Qwen 2.5 | negation | 0.0298 | 0.0397 | +0.0099 [+0.0081, +0.0123] |
| Qwen 2.5 | laterality | 0.0034 | 0.0043 | +0.0009 [+0.0007, +0.0012] |
| Qwen 2.5 | severity | 0.0117 | 0.0153 | +0.0035 [+0.0026, +0.0046] |
| Qwen 2.5 | anat. dir. | 0.0021 | 0.0027 | +0.0006 [+0.0004, +0.0008] |

## C   Dedicated paired experiment

The dedicated experiment isolates the effect of the $V$ term with a paired common-randomness design. For each model, a Standard and a $V$-regularised SAE are trained from identical initialisation, an identical reconstruction minibatch stream, and identical data order, differing only in the presence of the $V$ term with $\lambda_2 = 0.2$ on the negation family. Three outcomes summarise the run. First, the phenomenon is isolated, since the trained family's response inequality collapses from 0.233 to 0.0009 on GPT-2 and from 0.113 to 0.0008 on Qwen 2.5, a reduction of about 99 percent in both cases. Second, general reconstruction is unharmed, since explained variance on held OpenWebText tokens is unchanged at 0.9997 on GPT-2 and rises from 0.9950 to 0.9958 on Qwen 2.5. Third, the run exposes the saturation boundary reported in the scope paragraph of §6, since the Standard Qwen code already separates the negation family at an AUROC of 1.00 and the focused single-family objective reduces held-out linear separability to 0.81. The dedicated experiment therefore serves as a phenomenon and no-harm isolation, while the claim-carrying downstream evidence comes from the endpoints of §5.4, where the Standard code has headroom.

## D   Controlled non-saturated experiment

The synthetic experiment constructs a regime in which the task distinction is present in the input representation but not saturated in the Standard code, so that headroom exists by design. Inputs are drawn from a linear generative model in which a designated task direction carries a controlled fraction of the input variance, governed by a strength parameter $\alpha$, and paired Standard and $V$-regularised SAEs are trained with common randomness as in Appendix C. A fixed, rule-based operating point selects the regime before evaluation, with the rule that the Standard held-out probe sits below an AUROC of 0.95 while input-level recoverability stays above 0.99.

At the selected point the $V$-regularised code gives a held-out probe AUROC gain of +0.016 over the Standard code, with a seed-level standard error of comparable size, together with a lower-tail displacement gain $\Delta L_{20}(\|\Delta z\|)$ of +0.51 and a reconstruction ratio within 9 percent of the Standard run. Sweeping $\alpha$ toward stronger task signal drives the Standard probe to ceiling and the probe gain to zero, reproducing in a controlled setting the headroom dependence observed on the real models. The experiment is reported as a mechanism check rather than as a headline result, since the probe gain is modest, and it is read together with the real-model endpoints of §5.4.

## E   Held-out family generalisation audit

The audit tests whether the lower-tail lift is specific to the sixteen training families or a property of the learned basis. The evaluation draws on a pre-constructed pool of one hundred perturbation families with thirty fixed pairs each. The pool metadata records a creation date of 25 May 2026, and the held-out status is checkable from the released materials independently of any timestamp, since the eighty-four held-out families

appear in no training script, cache manifest, or reported training result of this paper. Sixteen entries of the pool are the training families themselves, and the remaining eighty-four are template-generated held-out families. The frozen general checkpoints are evaluated zero-shot on this held-out part, comprising two tiers. Tier A contains unseen instances of seen family kinds, and Tier B contains eighteen family kinds with no mechanical overlap with any training family, including unit reformatting, temporal shifts, dose-route changes, and word-order alternations. Families are subsampled to thirty pairs with per-family fixed seeds, and intervals are pair-level bootstrap intervals.

Six template families are exact duplicates of another pool entry and are excluded. Seven negation shards in the pool share their weakest pairs, so their lower-tail statistics coincide, and they are counted as one family in the macro summaries. After these exclusions, Tier A comprises fifty-four families. The lower-tail lift generalises in both tiers. On Tier A the macro mean $\Delta L_{20}$ is +0.016 on GPT-2, +0.036 on Qwen 2.5, and +0.026 on Gemma 2, with the interval excluding zero for 91, 89, and 100 percent of families respectively, and with every Gemma 2 family individually positive. On Tier B, the structurally novel tier, the macro mean is +0.026 on GPT-2, +0.040 on Qwen 2.5, and +0.033 on Gemma 2, with every GPT-2 and Gemma 2 family individually interval-positive. The families that fail to reach interval positivity concentrate where the scope paragraph of §6 predicts, on Qwen 2.5 surface-numeric shards. In this out-of-domain perturbation set the $V$-regularised code shows a higher perturbation-pair reconstruction error, most pronounced on Qwen 2.5, consistent with the reconstruction cost reported for the real-clinical domain, while general OpenWebText reconstruction is unaffected.

## F    Dose-value probe and the surface-token ceiling

This appendix supports the dose-value discussion in the scope paragraph of §6. A linear probe reads the numeric dose value from each representation on a held test split. In the hidden state the value is fully decodable, at an AUROC of 1.00 on all three models, and the Standard SAE code preserves this, also at 1.00. The $V$-regularised code is slightly worse on the value-distance endpoint, with a test $\Delta$AUROC of $-0.099$ on GPT-2, $-0.039$ on Gemma 2, and $-0.156$ on Qwen 2.5. The reading is that the digit sits on the surface of the text and is linearly decodable regardless of the SAE objective, so the dose-value distance probe has no headroom and is not the informative downstream test. The representation-level weak tail on the numeric family is lifted by the objective in the same way as the other families, as the held-out family audit shows (Appendix E); the ceiling is a property of the readout, not of the code response.

## G    Numeric orientation stress test

The stress test asks whether a non-saturated numeric regime can be found on the strongest model, so that a focused objective would have room to act. On Qwen 2.5 we score raw next-token log-probability margins on numeric minimal pairs, split into an easy regime and a trap regime designed to require the numeric distinction. The model handles both without a clean weak regime. Accuracy is 0.58 on the easy split and 0.87 on the trap split, and the lower-tail correct-margin remains near or above zero across families, so no clean non-saturated numeric regime is available on this model for a focused intervention. This null informs the choice, in the controlled experiment of Appendix D, to construct headroom synthetically rather than search for it in a saturated numeric task.

## H    Severity headroom preflight

Before committing a focused severity family, we measured the headroom available in each model on a severity-change probe with per-seed resampling. The hidden state already separates severity at an AUROC of 0.92 on GPT-2, 1.00 on Gemma 2, and 0.97 on Qwen 2.5, and the Standard SAE code is at or near this level. A focused $V$-regularised code does not improve on it, with a gate $\Delta$AUROC of $-0.156$ on GPT-2, $-0.061$ on Gemma 2, and $-0.017$ on Qwen 2.5. The preflight therefore predicts, before any focused training, that severity is a saturated target on frozen general checkpoints, which is consistent with the general finding that gains concentrate where the Standard code has headroom.

## I  Certificate-gated laterality repair

As a proof of mechanism we test whether a targeted repair, applied only to a certified laterality direction, changes downstream separation on the external OpenI left/right minimal pairs. On the same report-clustered protocol as the external boundary check, the zero-shot $V$-regularised code and a targeted-repair code give small report-cluster deltas with intervals that include zero, for example a lower-tail probability-margin delta of $+0.0022$ with interval $[-0.035, +0.030]$ for the zero-shot code on GPT-2. The repair moves the intended direction without a measurable downstream gain on this saturated external task, which is consistent with the boundary reading in §5.4. The experiment is reported as a mechanism probe rather than as evidence of downstream improvement.

## J  Multi-factor SAE architecture

The multi-factor SAE mirrors the multi-check-digit construction $D_n^m$ from the error-detecting-code framework (Balogh, 2026), in which $m$ independent check functions over an alphabet of size $n$ jointly detect a wider class of errors than any single function. It consists of $m$ parallel encoders, each mapping the hidden state $h \in \mathbb{R}^{d_{\text{in}}}$ to a factor code $z^{(k)} \in \mathbb{R}^{d_{\text{factor}}}$, and a shared decoder mapping the concatenation $z = [z^{(1)}; \ldots; z^{(m)}]$ back to $\hat{h}$. Total capacity is held constant at $d_{\text{factor}} \times m = d_{\text{sae}}$, so the comparison between $m{=}1$ and $m{>}1$ isolates the effect of factorisation at fixed capacity. Families are assigned to factors round-robin ($i \bmod m$); the $V$ loss for factor $k$ is computed on $z^{(k)}$ only, while evaluation computes $V_{\text{Gini}}$ on the concatenated code. In the six-family descriptive analysis, a single encoder regularised on all families reaches the low-imbalance regime. The architecture was tested at $m = 1, 2, 3, 6$ to confirm the $m{=}1$ result is not an artefact of the specific factorisation. (An orthogonality penalty $\lambda_{\text{ortho}} \|W^{(j)^\top} W^{(k)}\|_F^2$ encourages the factors to span complementary subspaces.)

## K  Single-run objective profiles

The single-run family-level comparison below supports the multi-seed baseline extension in Table 6. It is retained as numerical detail because it reports every evaluated perturbation family and includes the input-normalised $V$ variant.

## L  Mean sensitivity control

The per-family mean sensitivity values below check that the low-imbalance solution is not obtained by suppressing all perturbation responses.

## M  Input-normalised gain and perturbation complexity

The response $D_i$ in Eq. 3 does not account for the magnitude of the input perturbation. The input-normalised gain divides the feature-space response by the hidden-state response,

$$g_i = \frac{\|z_i^{\text{pert}} - z_i^{\text{orig}}\|/\|z_i^{\text{orig}}\|}{\|h_i^{\text{pert}} - h_i^{\text{orig}}\|/\|h_i^{\text{orig}}\|}, \tag{12}$$

and $V_{\text{Gini}}$ of the $g_i$ measures whether the SAE applies a similar amplification factor across all pairs under this normalisation. The V-reg$_n$ variant (trained with $\lambda_2 \cdot V_{\text{gain}}$) achieves $V_{\text{gain}}$ values that round to 0.000 on all six families on all three models (Table 12), indicating very low gain imbalance after input-complexity normalisation. The trade-off is expected. The two variants optimise different objectives, with V-reg targeting the raw response profile $D$ and V-reg$_n$ targeting the input-normalised profile $g$, so each minimises its own target and not the other. Thus, V-reg$_n$ does not minimise the raw $V$ (its raw values are comparable to or higher than the standard SAE on some families, Table 10). The two variants occupy complementary points on the Pareto surface of perturbation awareness, and the choice between them depends on whether raw or complexity-normalised sensitivity is the quantity of interest.

Table 10: $V_{\text{Gini}}$ (raw) across five SAE methods and three models (5,000 steps, $d_{\text{sae}} = 4{,}096$, 50 pairs per family). Bold indicates the lowest $V$ per family per model. The standard, JumpReLU, and MDL SAEs produce similar single-run profiles; the $V$-regularised SAE reduces response-profile imbalance in this descriptive comparison.

| Model | Family | Std | Jump | MDL | V-reg | V-reg$_\text{n}$ |
|---|---|---|---|---|---|---|
| GPT-2 | Negation | .219 | .222 | .223 | **.000** | .267 |
| | Typo | .307 | .306 | .309 | **.014** | .295 |
| | Synonym | .307 | .310 | .310 | **.018** | .326 |
| | Number swap | .264 | .265 | .266 | **.034** | .276 |
| | Semantic sub. | .435 | .433 | .435 | **.045** | .400 |
| | Word order | .281 | .282 | .282 | **.044** | .302 |
| Gemma 2 | Negation | .107 | .106 | .111 | **.000** | .188 |
| | Typo | .226 | .214 | .219 | **.000** | .308 |
| | Synonym | .214 | .215 | .219 | **.000** | .241 |
| | Number swap | .305 | .301 | .304 | **.156** | .469 |
| | Semantic sub. | .381 | .375 | .382 | **.290** | .468 |
| | Word order | .226 | .213 | .220 | **.000** | .189 |
| Qwen 2.5 | Negation | .136 | .131 | .133 | **.000** | .149 |
| | Typo | .160 | .158 | .162 | **.000** | .144 |
| | Synonym | .234 | .220 | .220 | **.002** | .177 |
| | Number swap | .304 | .308 | .306 | **.005** | .218 |
| | Semantic sub. | .369 | .367 | .370 | **.001** | .373 |
| | Word order | .105 | .103 | .098 | **.000** | .100 |

Table 11: Mean relative sensitivity per family (standard vs $V$-regularised SAE). The $V$-regularisation increases mean sensitivity on most families across all three models, confirming that it redistributes perturbation signals more uniformly rather than suppressing them.

| Model | Family | Standard | V-reg | Change |
|---|---|---|---|---|
| GPT-2 | Negation | 0.073 | **0.115** | +58% |
| | Typo | 0.067 | **0.081** | +21% |
| | Synonym | 0.043 | **0.050** | +17% |
| | Number swap | 0.021 | 0.023 | +7% |
| | Semantic sub. | 0.045 | 0.045 | 0% |
| | Word order | 0.043 | **0.057** | +34% |
| Gemma 2 | Negation | 0.342 | **0.614** | +79% |
| | Typo | 0.212 | **0.676** | +219% |
| | Synonym | 0.142 | **0.215** | +52% |
| | Number swap | 0.093 | **0.236** | +154% |
| | Semantic sub. | 0.215 | **0.257** | +20% |
| | Word order | 0.224 | **0.291** | +30% |
| Qwen 2.5 | Negation | 0.379 | **0.507** | +34% |
| | Typo | 0.245 | **0.289** | +18% |
| | Synonym | 0.199 | **0.213** | +7% |
| | Number swap | 0.111 | **0.134** | +21% |
| | Semantic sub. | 0.269 | **0.290** | +8% |
| | Word order | 0.302 | **0.358** | +18% |

The sixteen families differ in input-level complexity, measured by the coefficient of variation (CV) of hidden-state perturbation magnitudes and by centered effective rank (Table 13). The full geometry profile spans low-rank templatic families and more diffuse medical-domain families. The fact that V-reg reaches a low-

Table 12: $V_{\text{Gini}}$ (gain) for the V-reg$_n$ variant. Values are rounded to three decimals and indicate very low gain imbalance under input-complexity normalisation.

| Family | GPT-2 | Gemma 2 | Qwen 2.5 |
|---|---|---|---|
| Negation | 0.000 | 0.000 | 0.000 |
| Typo | 0.000 | 0.000 | 0.000 |
| Synonym | 0.000 | 0.000 | 0.000 |
| Number swap | 0.000 | 0.000 | 0.000 |
| Semantic sub. | 0.000 | 0.000 | 0.000 |
| Word order | 0.000 | 0.000 | 0.000 |

imbalance regime despite this heterogeneity shows that it is not merely benefiting from a homogeneous perturbation set.

Table 13: Input perturbation complexity per family. Each entry reports CV/centered effective rank for hidden-state perturbation magnitudes.

| Family | GPT-2 | Gemma 2 | Qwen 2.5 |
|---|---|---|---|
| Anatomical dir. | 0.61/4.1 | 0.50/17.4 | 0.48/18.6 |
| Body part swap | 0.60/3.8 | 0.60/13.8 | 0.58/14.6 |
| Causal reversal | 0.70/5.2 | 0.50/18.9 | 0.65/16.6 |
| Condition neg. | 0.73/2.1 | 0.45/17.6 | 0.61/12.1 |
| Date/time change | 0.71/3.1 | 0.52/18.2 | 0.66/14.8 |
| Drug name swap | 0.56/4.1 | 0.32/23.0 | 0.46/18.5 |
| Frequency change | 1.13/2.6 | 0.63/14.6 | 0.71/12.6 |
| Gender swap | 0.83/2.7 | 0.29/18.1 | 0.21/20.7 |
| Negation | 0.52/2.4 | 0.20/16.3 | 0.26/15.7 |
| Number swap | 0.54/3.7 | 0.54/21.8 | 0.44/26.2 |
| Semantic sub. | 0.89/5.2 | 0.69/19.9 | 0.90/12.6 |
| Severity change | 0.85/2.9 | 0.54/18.2 | 0.69/15.3 |
| Synonym | 0.65/2.6 | 0.27/36.7 | 0.33/34.4 |
| Typo | 0.61/5.1 | 0.27/36.6 | 0.29/35.8 |
| Unit measure | 0.61/4.1 | 0.60/13.4 | 0.72/10.8 |
| Word order | 0.67/2.4 | 0.40/28.5 | 0.21/35.7 |

## N  Reconstruction quality across methods

On GPT-2, the V-reg MSE (0.0069) is about twice the standard (0.0034), a modest cost for a $10\times$ reduction in $V$. On Gemma 2, V-reg achieves the lowest MSE of all five methods (0.0119 vs standard 0.0130), indicating that perturbation-aware orientation can sometimes improve reconstruction. On Qwen 2.5, V-reg has the highest MSE (0.0104), but the absolute level remains low. The V-reg$_n$ variant consistently achieves reconstruction comparable to or better than the standard, because the gain-normalised $V$ loss places less pressure on the encoder weights.

## O  Hidden-state extraction protocol

Decoder-only language models can use different padding conventions during batched evaluation. To compare perturbation responses at matched sequence positions, all hidden-state caches in the evaluation use a true-last-token extraction rule. For each sequence, the extraction index is the final position with attention-mask value one. This rule selects the last non-padding token independently of whether the tokenizer pads on the left or on the right.

Table 14: Reconstruction MSE and sparsity (fraction of active features) across five SAE methods and three models. All share $d_{\text{sae}} = 4{,}096$ and 5,000 steps.

| Model | Method | MSE | Frac active |
|---|---|---|---|
| GPT-2 | Standard | 0.0034 | 0.352 |
| | JumpReLU | 0.0035 | 0.333 |
| | MDL | 0.0065 | 0.337 |
| | V-reg | 0.0069 | 0.344 |
| | V-reg$_n$ | 0.0033 | 0.329 |
| Gemma 2 | Standard | 0.0130 | 0.072 |
| | JumpReLU | 0.0222 | 0.097 |
| | MDL | 0.0250 | 0.076 |
| | V-reg | 0.0119 | 0.094 |
| | V-reg$_n$ | 0.0142 | 0.071 |
| Qwen 2.5 | Standard | 0.0052 | 0.172 |
| | JumpReLU | 0.0020 | 0.204 |
| | MDL | 0.0038 | 0.248 |
| | V-reg | 0.0104 | 0.162 |
| | V-reg$_n$ | 0.0040 | 0.160 |

This convention is used for GPT-2, Gemma 2, and Qwen 2.5 perturbation caches. It makes the response quantity in Eq. (3) well-defined across models, because $z_i^{\text{orig}}$ and $z_i^{\text{pert}}$ are computed from the same semantic sequence position in the original and perturbed inputs. The same cached hidden states are then reused for Standard, JumpReLU, MDL, and $V$-regularised SAE evaluation under the frozen multi-seed protocol.

## P   Reporting rounded small values

Several tables display $V_{\text{Gini}}$ values rounded to three decimals. An entry printed as 0.000 should be read as a rounded small value, not as a mathematical claim that the underlying response or imbalance is exactly zero. The full-precision values are retained in the machine-readable result files. This convention is especially important for low-imbalance summaries. A rounded value indicates that the response profile is close to uniform at the displayed precision; the scale-sensitive quantities in §2 report response magnitude. Two factors make such values more likely. The $\varepsilon$ in Eq. (3) stabilises small denominators and compresses very small responses toward a common scale, and for small or internally similar families the sorted response profile can approach uniformity, which drives the Gini statistic below the displayed precision. Near-zero cells are therefore accompanied by seed dispersion (Table 2) and by the family-geometry controls of Appendix Q, so that a near-zero $V_{\text{Gini}}$ is not read as a trivial saturation effect.

## Q   Family-geometry and template-cluster controls

The perturbation set contains repeated surface templates, so uncertainty should not rely only on pair-level resampling. We infer exact-skeleton template clusters by replacing the changed span in each original/perturbed pair with a slot marker. Across the 600 perturbation pairs, this procedure finds 424 template clusters, of which 350 are singleton templates and 74 are multi-pair templates. In total, 250/600 pairs (41.7%) belong to repeated templates. Repeated exact surface templates occur in five families, namely `negation`, `number_swap`, `semantic_substitution`, `synonym`, and `typo`. The other eleven families are exact-skeleton singleton families, so template-cluster resampling coincides with pair-level resampling for those families.

Table 15 summarises the main resampling checks. The fixed-family intervals resample template clusters within each family while keeping the sixteen evaluated families fixed. The family-resampled intervals additionally resample families. The latter are wider, as expected, but remain positive for the relative lower-tail effect and for the absolute code-displacement lower tail.

Table 15: Template-cluster resampling summary. LOTO denotes leave-one-template-out, and LOFO denotes leave-one-family-out.

| Model | LOTO positive templates | LOFO $\Delta L_{20}(D)$ range | Centered effective-rank range |
|---|---|---|---|
| Gemma 2 | 424/424 | +0.1149–+0.1401 | 13.4–36.7 |
| GPT-2 | 424/424 | +0.0522–+0.0584 | 2.1–5.2 |
| Qwen 2.5 | 424/424 | +0.1140–+0.1372 | 10.8–35.8 |

The hidden-state perturbation geometry differs across models. GPT-2 has a lower-dimensional and more direction-concentrated perturbation geometry than Gemma 2 or Qwen 2.5, but its sparse-code lower-tail response still improves under $V$-regularisation. Family-size ablations also support the same conclusion. At $K = 10$ pairs per family, all relative lower-tail effects remain positive in all repeats for Gemma 2 and GPT-2, and the worst Qwen 2.5 family is positive in 99.9% of repeats. Absolute code-distance effects are more heterogeneous at small $K$, especially for Qwen 2.5, but the aggregate cluster-aware intervals remain positive. Decoded-response lower-tail effects are stable for Gemma 2 and Qwen 2.5, while GPT-2 remains near zero after decoding.

## R  Sixteen-family per-family values

The per-family values below provide the detailed support for the sixteen-family summary in Table 8.

Table 16: Per-family $V_{\text{Gini}}$ (raw) for the sixteen-family joint $V$-regularised SAE on GPT-2 and Qwen 2.5, standard SAE versus V-reg. The six families above the rule are the original families; the ten below are the auto-generated medical-domain families, none of which informed the metric or training protocol. Both groups reach a low-imbalance regime in this descriptive comparison. Source files `results/joint_eval_{gpt2,qwen-2.5-3b}.json`.

| | GPT-2 small | | Qwen 2.5 3B | |
|---|---|---|---|---|
| Family | Std | V-reg | Std | V-reg |
| Negation | 0.219 | **0.000** | 0.136 | **0.000** |
| Typo | 0.307 | **0.000** | 0.160 | **0.000** |
| Synonym | 0.307 | **0.000** | 0.234 | **0.000** |
| Number swap | 0.264 | **0.000** | 0.303 | **0.000** |
| Semantic substitution | 0.435 | **0.035** | 0.368 | **0.000** |
| Word order | 0.281 | **0.001** | 0.105 | **0.000** |
| *6-family mean* | *0.302* | *0.006* | *0.218* | *0.000* |
| Anatomical direction | 0.279 | **0.000** | 0.225 | **0.000** |
| Body-part swap | 0.238 | **0.000** | 0.258 | **0.000** |
| Causal reversal | 0.352 | **0.000** | 0.291 | **0.000** |
| Condition negation | 0.294 | **0.000** | 0.192 | **0.000** |
| Date/time change | 0.350 | **0.000** | 0.294 | **0.000** |
| Drug-name swap | 0.266 | **0.000** | 0.183 | **0.000** |
| Frequency change | 0.377 | **0.000** | 0.341 | **0.000** |
| Gender swap | 0.337 | **0.000** | 0.111 | **0.000** |
| Severity change | 0.428 | **0.000** | 0.320 | **0.000** |
| Unit of measure | 0.296 | **0.000** | 0.299 | **0.000** |
| *10-family medical-domain mean* | *0.322* | *0.000* | *0.251* | *0.000* |
| **All-16 mean** | **0.314** | **0.002** | **0.239** | **0.000** |

## S   Gemma extraction consistency

Decoder-only models differ in their padding conventions, and under Gemma's left-padding convention a fixed-index extraction rule selects a padding position rather than the final token, so caches built with such a rule are not computed at the true last token. All Gemma 2 analyses in this paper therefore use the attention-mask-based true-last-token rule of Appendix O, the same rule applied to GPT-2 and Qwen 2.5. Under this rule Gemma follows the same qualitative pattern as the other two models in the primary response-profile metrics. An apparent imbalance plateau that arises under fixed-index extraction on Gemma is an artefact of the misaligned token position rather than a structural property of the Gemma hidden state, which is why the position-level rule matters for cross-model comparability. The scientific interpretation is based on the true-last-token caches and the frozen cross-model evaluation reported in Table 2.

## T   Pre-caching of perturbation hidden states

The $V$ term operates on deterministic, pre-cached perturbation hidden states. Each perturbation pair is passed through the frozen language model once, stored, and reused at every SAE training step. This mirrors standard activation caching for the reconstruction loss and ensures that changes in the perturbation-response objective come from SAE optimisation rather than from stochastic variation in language-model forward passes. For production use, $\lambda_2$ should be selected on a validation set via the Pareto frontier of reconstruction MSE versus mean $V_{\mathrm{Gini}}$.

## U   Perturbation family descriptions

The six original families contain 50 sentence pairs each; the ten medical-domain families (used in the sixteen-family study of §5.11) contain 30 pairs each, for 600 pairs in total. Representative examples for every family follow; the complete datasets are provided as supplementary material (see Reproducibility and supplementary material).

### U.1   Original families

**Semantic substitution.**   "The cat sat on the mat..." → "The dog sat on the mat..."; "...very hot and sunny..." → "...very cold and sunny..."; "...drove her car..." → "...drove her truck..."; "The doctor examined..." → "The nurse examined..."; "...from the small shop" → "...from the small store".

**Typo (adjacent-character transposition).**   "...admitted to the hospital..." → "...hopsital..."; "...published the results..." → "...publsihed..."; "...an important message..." → "...mesasge..."; "...their assignment..." → "...thier..."; "...explained the concept..." → "...explaind...".

**Negation.**   "The test result is positive..." → "...is not positive..."; "The drug is safe..." → "...is not safe..."; "The system is working..." → "...is not working..."; "The patient is responding..." → "...is not responding..."; "The data is consistent..." → "...is not consistent...".

**Synonym.**   "...very big..." → "...very large..."; "...happy to hear..." → "...glad to hear..."; "...very hard to complete..." → "...very difficult..."; "She started working..." → "She began working..."; "...fix the broken machine..." → "...repair...".

**Number swap.**   "...received 3 doses..." → "...8 doses..."; "...5 hospitals..." → "...9 hospitals..."; "...2 pills..." → "...7 pills..."; "...scored 90 points..." → "...40 points..."; "...reached 30 degrees..." → "...80 degrees...".

**Word order (adjacent-word swap).**   "The black cat..." → "The cat black..."; "She quickly finished..." → "She finished quickly..."; "The old man..." → "The man old..."; "The bright sun..." → "The sun bright..."; "The young student..." → "The student young...".

## U.2   Additional medical-domain families

These ten auto-generated families were not used to design the metric or training protocol; the sixteen-family study (§5.11, Table 16) evaluates generalisation to them. Three representative pairs per family follow.

**Anatomical direction (anterior↔posterior, proximal↔distal, dorsal↔ventral).**   "The mass is located in the *anterior* mediastinum" → "...*posterior* mediastinum"; "Pain is felt in the *proximal* part of forearm" → "...*distal* part..."; "The lesion is on the *dorsal* surface of hand" → "...*ventral* surface...".

**Body-part swap (left↔right).**   "...pain in the *left* knee" → "...*right* knee"; "Fracture detected in the *right* femur" → "...*left* femur"; "Swelling observed in the *left* ankle" → "...*right* ankle".

**Causal reversal.**   "The fever was caused by the infection" → "The infection was caused by the fever"; "Pain *increased* after taking the medication" → "Pain *decreased*..."; "...resulted in significant *improvement*" → "...significant *deterioration*".

**Condition negation.**   "The patient *has* a history of diabetes" → "...*has no* history of diabetes"; "Allergies to penicillin *are* documented" → "*No* allergies...documented"; "There *is* evidence of metastatic disease" → "There *is no* evidence...".

**Date/time change.**   "...scheduled for *Monday* morning" → "...*Friday* morning"; "...admitted on *January* 15" → "...*March* 15"; "Follow-up in *2* weeks" → "...*6* weeks".

**Drug-name swap.**   "...prescribed *metformin* daily" → "...*lisinopril* daily"; "Start *ibuprofen* 400 mg..." → "Start *acetaminophen* 400 mg..."; "...taking *warfarin* for clots" → "...*aspirin* for clots".

**Frequency change.**   "Take the medication *once* daily" → "...*three times* daily"; "Apply the cream *twice* daily" → "...*once weekly*"; "...given every *two weeks*" → "...every *two months*".

**Gender swap (he↔she, his↔her).**   "...reported *he* felt dizzy" → "...*she* felt dizzy"; "*She* was admitted to the ward" → "*He* was admitted..."; "*His* blood pressure was elevated" → "*Her* blood pressure...".

**Severity change (mild→severe, minor→major).**   "...*mild* chest pain" → "...*severe* chest pain"; "...*moderate* difficulty breathing" → "...*extreme* difficulty..."; "...*slight* swelling" → "...*significant* swelling".

**Unit of measure (mg→g, mL→L, cm→mm).**   "The dosage is 500 *mg*..." → "...500 *g*..."; "Inject 0.5 *mL*..." → "Inject 0.5 *L*..."; "...measured 2.3 *cm*..." → "...2.3 *mm*...".

