# OpenReview forum: "A symmetry-matching approach to blind-spot reduction in sparse autoencoders"
_TMLR — Under review for TMLR_

### Review · Reviewer_tPEL · 2026-06-19

**Summary Of Contributions:**

The paper studies "representational blind spots" in language models: the phenomenon that semantically distinct inputs (e.g., "3 doses" vs. "8 doses") can map to near-interchangeable deep representations. The authors propose V-Gini, the Gini coefficient of feature-space sensitivities across a family of input perturbations, as a differentiable, family-specific measure, and add it as a regularisation term to the SAE training loss (with pre-cached perturbation hidden states for efficiency). They compare against standard L1, JumpReLU, and MDL objectives on GPT-2 small, Gemma 2 2B, and Qwen 2.5 3B.

The main empirical claims are:

- (i) Standard, JumpReLU, and MDL objectives have near-identical V-Gini profiles and do not reduce it, while V-regularisation reduces it by an order of magnitude.
- (ii) A single jointly regularised basis suffices for all families simultaneously (m* = 1).
- (iii) The result generalises from 6 to 16 perturbation families, including 10 held-out auto-generated medical families, reaching mean V close to 0 on GPT-2 and Qwen.
- (iv) Gemma 2 layer 13 exhibits a structural floor (V close to 0.15) invariant to a 20x hyperparameter sweep, interpreted as a limit of the hidden state rather than of the objective.

**Key strengths.** The Gemma layer-13 analysis (§5.8, Appendix E) is careful and honest: an 18-configuration one-axis sweep plus bfloat16/activation-clipping controls, with the limitation localised to the hidden state rather than the objective. The reconstruction-cost trade-offs (Appendix D) are transparently reported, and the Broader Impact statement is appropriately cautious about the limits of the method.

**Key weaknesses.** The paper's central object of study, a "blind spot", is never formally defined, and the paper in fact uses several non-equivalent notions of it, none of which matches the quantity V-Gini actually computes. This definitional gap undermines the interpretation of the headline results. Compounding it, there is no downstream-task evaluation linking low V to improved discriminability, no reported variance across seeds, and the auto-generated perturbation families are small and possibly internally homogeneous in ways that make low V easy to attain.

**Audience:**

Yes

**Audience Explanation:**

**Yes.** The intersection of sparse-autoencoder interpretability and safety-relevant robustness is of clear interest to a substantial subset of the TMLR audience. The observation that reconstruction, sparsity, and description-length objectives are essentially blind to perturbation-uniformity, the m* = 1 result, and especially the carefully characterised Gemma layer-13 floor are findings researchers in interpretability would learn from, independent of the framing concerns above.

**Broader Impact Concerns:**

A Broader Impact Statement is present and, on balance, appropriately cautious: it notes that low V on a tested family does not guarantee safety on untested families, that the Gemma floor shows the method cannot manufacture information a model does not encode, and that the perturbation sets are synthetic and English-only.

My concern is that the main text (Discussion §6) suggests regulators could specify V thresholds and developers demonstrate compliance via V profiles. Given the definitional issues above, in particular that V-Gini is scale-invariant and so does not measure absolute detectability, promoting V as a compliance or certification signal is premature and could give false assurance: a system uniformly insensitive to a safety-relevant family would pass a V threshold. I would ask the authors to add a sentence to the Broader Impact statement stating explicitly that V-Gini measures within-family uniformity of sensitivity rather than absolute detectability, and that it should not be used as a standalone certification signal without an absolute-sensitivity or downstream validation.

**Claims And Evidence:**

No

**Claims Explanation:**

**No, in the current form.** The gap is addressable, but it begins with a definitional problem that must be fixed before the evidence can be evaluated at all.

**The central term is undefined and used inconsistently.** "Blind spot" carries the paper's entire argument yet is never formally defined, and the text slides between at least three distinct, non-equivalent notions:

1. **Algebraic** (inherited from the error-detecting-code framing): a perturbation in the stabiliser, i.e., one that leaves the encoding *exactly* unchanged, F(g.x) = F(x). Binary and exact; its "size" is a count of invisible family members, and it is what the condition Stab(G,F) intersect E = {e} formalises.
2. **Representational** (what the motivation actually evokes): *low absolute sensitivity* of the deep representation to a change that matters. The dosage example is alarming precisely because the representation barely moves when 3 becomes 8. This notion is continuous and magnitude-based.
3. **Operationalised** (what every number in the paper measures): the Gini coefficient of the per-pair sensitivities *within a family*, i.e., the inequality/dispersion of the s_i.

These come apart provably, because the Gini coefficient is **scale-invariant**: multiplying every s_i by a constant leaves V-Gini unchanged, so V-Gini carries no information about whether sensitivities are large or small in absolute terms. Two consequences follow directly, both at odds with reading V as "blind-spot severity":

- A family in which every pair has sensitivity around 0.001, meaning the representation is essentially blind to all of them and so a maximal blind spot in sense (2), has V-Gini = 0, which the paper would report as "no blind spot."
- A family with uniformly high sensitivities except one slightly higher pair has V-Gini > 0 and would be flagged.

The persuasive force of the paper relies on the reader holding the sense-(2) intuition while the evidence concerns only sense (3): that within-family sensitivities became more equal to one another. The §5.5 / Table 3 result that mean sensitivity rises is a useful partial guard against the pure-suppression failure mode, but it does not close this gap, and the absolute magnitudes remain small with no task-justified standard for what is sufficient.

There is also an **internal inconsistency between the announced condition and the metric**. For a continuous SAE encoder, distinct hidden states essentially never produce identical codes, so s_i > 0 almost surely and the exact stabiliser is already {e} with probability one, whereas V-Gini = 0 requires all s_i *equal*, a strictly different and much stronger condition. The principle the paper states (Stab(G,F) intersect E = {e}) is therefore not the condition its metric checks, and that bridge is never built. The one verbal gloss that is coherent, namely numeric perturbations showing "near-synonym-level insensitivity" (a *between-family* comparison), is not what V-Gini measures either, since it is computed strictly within a single family and never compares, e.g., number-swap sensitivity against synonym sensitivity. The only operational stand-in for "has a blind spot" is the V = 0.05 line in the figures and the "V < 0.05" / "V < 0.005" counts, but 0.05 is a threshold on a scale-invariant Gini coefficient and is never justified as a meaningful detectability boundary.

**No downstream evidence links the metric to the safety claim.** The motivation is safety-critical: a representation that conflates dosages propagates the error downstream. Yet no experiment shows that a low-V SAE actually distinguishes such inputs better on any probing, classification, or downstream task. The link from "low V" to "safer downstream behaviour" is asserted, not demonstrated.

**The core comparison risks circularity.** "Only the objective optimising V reduces V, while JumpReLU/MDL do not" is close to tautological: a regulariser for X reduces X. The informative claim would be that reducing V improves an *independent* quantity, which is exactly what is missing.

**No variance is reported.** I see no multi-seed results, while the paper reports many exact V = 0.000 values and precise reductions (-99.3%) and calls differences below 0.02 among Standard/JumpReLU/MDL "nearly identical." Without variance these are not convincingly established.

**The generalisation control is incomplete.** The 16-family result rests on small (30-pair), auto-generated, template-like families (Appendix G), several near structurally identical within a family. Internal homogeneity could make within-family uniformity (V close to 0) easy independent of any learned orientation, and the complexity statistics (CV, Table 7) are reported only for the six original families, not the ten medical ones, so the "generalisation, not overfit" argument lacks the control that would distinguish the two explanations.

To be clear, much of the underlying evidence *does* support narrower claims: V-regularisation reliably reduces within-family sensitivity inequality without much reconstruction cost, alternative objectives do not, and the Gemma floor is robust. The issue is the distance between those narrower claims and the safety-framed, blind-spot-elimination claims actually made, a distance that starts with the undefined central term.

**Requested Changes:**

1. **[Critical]** Define "blind spot" formally and use a single definition consistently. The paper conflates (a) the algebraic stabiliser notion (exact invisibility), (b) absolute insensitivity of the representation, and (c) within-family inequality of sensitivities, which is what V-Gini measures. These are provably non-equivalent: V-Gini is scale-invariant, so a uniformly-blind family scores V = 0 while a uniformly-visible family can score V > 0. Commit to one definition and demonstrate that V-Gini is a faithful proxy for *that* definition. If the intended notion is absolute insensitivity, the natural measure is the fraction of family members with sensitivity below a task-justified threshold tau (or mean shortfall below tau), and an absolute metric rather than a Gini coefficient is required. If the intended notion is detectability against the code's own noise floor, a signal-detection framing (e.g., AUC of perturbed-vs-identity sensitivities) would be appropriate. As written, V-Gini is best described as measuring *fairness of sensitivity across a family*, a defensible secondary desideratum but not "blind-spot severity," and the text should either adopt that honest description or supply the missing absolute/detectability measure.

2. **[Critical]** Reconcile the announced condition with the metric. Explain why Stab(G,F) intersect E = {e} (which holds almost surely for a continuous encoder, since s_i > 0) is the relevant principle when the metric instead targets equality of the s_i, and justify the V = 0.05 threshold (and the "V < 0.05" / "V < 0.005" counts) as corresponding to a meaningful detectability boundary rather than an arbitrary cut on a scale-invariant coefficient.

3. **[Critical]** Add at least one downstream or probing evaluation linking low V to improved discriminability on an independent task (e.g., a numeric-value or negation probe on the SAE codes). If such evidence is not added, downgrade the clinical/safety narrative to motivation rather than a demonstrated outcome throughout.

4. **[Critical]** Report variance across multiple random seeds for the headline comparisons (Tables 1, 2, 4, 5; Figure 1), including the exact-0.000 cells and the "nearly identical" claim among Standard/JumpReLU/MDL. State whether reported values are single-run or averaged.

5. **[Critical]** Provide the perturbation-complexity statistics (CV of hidden-state perturbation magnitude, as in Table 7) for all sixteen families, and add a control or analysis ruling out that the low V on the ten medical families is a consequence of within-family structural homogeneity rather than learned orientation.

6. **[Strengthening]** Clarify how much of the algebraic framework (the Stab(G,F) intersect E condition, the stabiliser monotonicity theorem, the (H,V,G) triad) is load-bearing for the method versus motivational, given that the method reduces to a Gini coefficient of sensitivities. Since the grounding rests on the Balogh 2026a/2026b preprints, a self-contained statement of the minimal results relied upon would help.

7. **[Strengthening]** Tighten the consistency of the headline percentages. The abstract's "83-100%" appears tied to the training-loss trajectory (Figure 2 caption), while the per-family evaluation reductions are reported separately; make explicit which quantity each percentage refers to and how it is computed.

8. **[Strengthening]** Discuss why exact V = 0.000 arises in so many cells (relation to the epsilon in Eq. 1 and to family size), and whether the metric saturates trivially for small or homogeneous families.

---

> ### Author Response · Authors · 2026-06-21
>
> We sincerely thank the reviewer for the detailed and insightful comments. After re-evaluating the manuscript, we concur that several points require substantive revision. We would like to briefly outline the planned changes before submitting the revised manuscript and full response.
>
> **1. Formal definition of blind spots**
>
> As the reviewer rightly noted, the submitted version did not define the central term precisely enough. We will distinguish exact zero-response blind cases, approximate low-response cases, and sensitivity-profile imbalance. In the SAE setting, we will define a perturbation response such as
>
> $$
> D_i = \frac{\lVert z_i^{\mathrm{pert}}-z_i^{\mathrm{orig}}\rVert}{\lVert z_i^{\mathrm{orig}}\rVert+\varepsilon},
> $$
>
> using the same small-$\varepsilon$ convention as in the original sensitivity definition. This gives
>
> $$
> S_0=\lbrace i \mid D_i=0 \rbrace,
> $$
>
> for exact zero-response cases,
>
> $$
> S_\gamma=\lbrace i \mid D_i<\gamma \rbrace,
> $$
>
> for approximate low-response cases, and
>
> $$
> V_{\mathrm{Gini}}=\operatorname{Gini}(D_1,\ldots,D_K)
> $$
>
> for sensitivity-profile imbalance. The revision will make clear that these are different objects. In practice, $S_0$ is typically empty for a continuous encoder, which is why $S_\gamma$ is the operational object.
>
> **2. Relation between the stabilizer condition and the metric**
>
> The exact condition
>
> $$
> \operatorname{Stab}(G,F)\cap E=\lbrace e \rbrace
> $$
>
> will be stated as the zero-response form of the blind-set principle. The experimental metric does not by itself test this exact algebraic condition. Instead, $V_{\mathrm{Gini}}$ measures inequality in the finite perturbation-response profile. The revision will make this bridge explicit. The blind-set condition defines the failure mode, approximate margins define operational detectability, and the Gini statistic measures finite-profile imbalance.
>
> **3. Scale invariance and absolute detectability**
>
> We agree that $V_{\mathrm{Gini}}$ is not an absolute detectability metric or safety certificate. It measures how unevenly perturbation sensitivity is distributed across a finite family. In the loss, it is a differentiable conditioning term that discourages concentration in a few directions. Detectability must be evaluated by scale-sensitive quantities. The revision will include absolute-sensitivity analyses using $\Delta L_{20}(D)$, $\Delta L_{20}(\lVert\Delta z\rVert)$, and decoded-response lower-tail metrics. These test whether V-reg raises weak responses rather than merely decreasing the optimized Gini statistic.
>
> **4. Thresholds and near-zero values**
>
> We share the reviewer's view that $V<0.05$ and $V<0.005$ should not be presented as detectability boundaries. If retained, they will be descriptive low-imbalance summaries only. Substantive detectability claims will be tied to response magnitude, lower-tail behavior, and independent validation. Values displayed as $0.000$ will be reported at full precision or described as rounded near-zero values, with numerical and family-size considerations made explicit.
>
> **5. Downstream and probe evidence**
>
> We recognize that downstream usefulness should be tested independently. The revision will include held-out and external probe evaluations to test whether response-profile changes correspond to improved discriminability beyond the metric itself. We will present these as validation evidence, not as a standalone safety guarantee.
>
> **6. Variance and baseline comparisons**
>
> Seed variance is indeed needed for the headline comparisons. The revised manuscript will report uncertainty across seeds for Standard versus V-reg. We will report the non-perturbation-aware baselines, including JumpReLU and MDL, under the same metric hierarchy.
>
> **7. Perturbation-family homogeneity**
>
> We share the concern that small or homogeneous perturbation families could make low imbalance easier to obtain. The revision will add family-geometry and family-size controls, including perturbation-complexity statistics for all families, so that low $V_{\mathrm{Gini}}$ is not interpreted as a trivial consequence of family construction.
>
> **8. Claim language and broader impact**
>
> Finally, we will reframe V-reg as a perturbation-sensitivity conditioning objective. The blind-set condition defines the failure mode, approximate margins define operational detectability, and the Gini term provides smooth training pressure that improves the finite response profile. We will revise threshold interpretation, exact-zero wording, percentage summaries, and broader-impact statements. The revised manuscript will state that $V_{\mathrm{Gini}}$ should not be used as a standalone certification signal without absolute-sensitivity or downstream validation.
>
> We appreciate the reviewer's constructive comments. The revised manuscript and accompanying point-by-point response will detail the corresponding definitions, tables, and uncertainty estimates.

---

> > ### Author Response · Authors · 2026-07-04
> > **Response part 2 of 4.**
> >
> > **3. Scale invariance and absolute detectability.** We agree that $V_{\mathrm{Gini}}$ is not an absolute detectability metric or a safety certificate. We now state its scale invariance formally, $V_{\mathrm{Gini}}(cD)=V_{\mathrm{Gini}}(D)$ for all $c>0$ (Eq. 9), and draw the direct consequence that it carries no information about absolute response magnitude. Detectability is therefore evaluated by scale-sensitive quantities. Table 3 reports $\Delta L_{20}(D)$, $\Delta L_{20}(\lVert\Delta z\rVert)$, decoded-response support, and input-normalised gain. These test whether $V$-regularisation raises weak responses rather than merely equalising the Gini statistic. The multi-seed results (Table 2) confirm that both relative and absolute lower-tail responses increase under $V$-reg.
> >
> > **4. Thresholds and near-zero values.** We agree that $V<0.05$ and $V<0.005$ should not be read as detectability boundaries. Where such counts are retained (Table 8) they are labelled as descriptive low-imbalance summaries only, and substantive claims are tied to response magnitude, lower-tail behaviour, and independent validation. Values that previously appeared as exact $0.000$ are now reported with seed dispersion (Table 2) or as rounded near-zero values; the convention and the effect of family size are stated in Appendix P.
> >
> > **5. Downstream and probe evidence.** We added three independent checks in Section 5.4, namely a held-out negation-template probe, a null-calibrated separation test against meaning-preserving nuisance controls (Table 5), and an external OpenI laterality probe. These enter the paper as validation evidence, not as a standalone safety guarantee. The held-out point estimates improve on all three models but the template-bootstrap intervals include zero, so the effect is suggestive rather than uniform. Under the null-calibrated test Gemma and Qwen show positive separation while GPT-2 is uncertain in point direction. The external OpenI laterality probe deltas are small and cross zero on GPT-2 and Qwen 2.5, while the Gemma 2 delta is negative with a report-clustered interval of $[-0.163, -0.012]$ that excludes zero, on this saturated external task. The summary, now stated in the text, is that $V$-regularisation improves the geometry of identified weak directions while transfer to downstream probability margins is task- and model-dependent. Beyond these, the revision adds a zero-shot real-clinical lower-tail audit on OpenI radiology minimal pairs (Table 9, Appendix B), where the lift appears in all eight model-by-family cells with report-clustered intervals excluding zero.
> >
> > We also address the related concern that "only the objective optimising $V$ reduces $V$" is close to tautological. The informative comparison is not that $V$-reg reduces $V$, but that the non-perturbation-aware objectives, which do not optimise $V$, leave the response profile essentially unchanged (Table 6), while $V$-reg additionally improves the independent probe separation reported above. The reduction is therefore tied to quantities the objective does not directly optimise, indeed, by scale invariance the objective cannot by construction raise the absolute tail magnitude, and on the real-clinical data the lift and the trained inequality statistic move independently.
> >
> > **6. Variance and baseline comparisons.** The headline comparisons are now reported under a frozen multi-seed protocol with mean and standard deviation across seeds for Standard versus $V$-regularised SAEs (Table 2). JumpReLU and MDL are reported under the same metric hierarchy and the same caches and seeds (Table 6), so the statement that the non-$V$ objectives track the Standard profile is now established with dispersion rather than from a single run.
> >
> > **7. Perturbation-family homogeneity.** We added family-geometry and family-size controls (Table 4) and perturbation-complexity statistics for all sixteen families (Table 13, Appendix M). A template-cluster analysis identifies repeated templates and resamples them, and a family-resampled bootstrap additionally resamples families. The effect remains positive under these controls, and $V$-regularisation also reduces imbalance on high-complexity families, so low $V_{\mathrm{Gini}}$ is not a trivial consequence of within-family homogeneity.
> >
> > *(Continued in part 3.)*

---

> ### Author Response · Authors · 2026-07-04
> **Response part 1 of 4.**
>
> We thank the reviewer again for the depth and precision of the review. The concern that the central term was under-defined was well founded, and acting on it substantively changed several parts of the manuscript. Working through the definitional separation the review called for also led us to re-audit the hidden-state extraction protocol, which in turn corrected the Gemma analysis. We describe the scope of the revision and that audit first, because they touch several of the itemised points, and then respond to each request in the order it was raised.
>
> **What changed and what is preserved.** Because the revision is substantial, we state its scope up front. The original experiments and results all remain in the manuscript; nothing has been replaced, only corrected where the extraction-protocol audit required it. The claims have been narrowed rather than redirected. The title now reads "reduction" rather than "elimination", a change that reflects the same narrowing that all three reviews converged on, and the blind-set condition continues to define the failure mode in the revised text. Every added analysis answers a named reviewer request; the main-text claim carriers are unchanged, the new supporting material is placed in appendices, and the boundary cases are reported alongside the positive results.
>
> **Extraction-protocol audit and the Gemma analysis.** The definitional separation the review required, between exact zero response, margin-based low response, and finite-profile imbalance, prompted us to recheck whether the hidden-state extraction protocol was aligned across models. The recheck showed that Gemma's left-padding convention affected the final-token position used by the original extraction rule; the misalignment was barely visible in the aggregate metrics, and only the exact-versus-margin distinction the review insisted on made a position-level re-audit necessary, so the correction is a direct product of the review's precision. We rebuilt the Gemma perturbation cache using a true-last-token extraction rule computed from the attention mask, and reran the relevant analyses under the corrected protocol. Under this protocol Gemma no longer behaves as an outlier in the primary response-profile metrics. It follows the same qualitative pattern as GPT-2 and Qwen, in that $V$-regularisation raises lower-tail relative response and lower-tail absolute code displacement while reducing sensitivity-profile imbalance. As a consequence, the revised manuscript no longer presents the earlier Gemma layer-13 "structural floor" as a primary model-limit result. Gemma is now included in the same corrected, frozen evaluation framework as the other models (Appendix S states the corrected rule and the artefact). The corrected protocol supersedes the original floor analysis, and the corrected result strengthens rather than weakens the central claim, since it removes a model-specific exception to the response-profile effect.
>
> **1. Formal definition of blind spots.** We agree the submitted version did not define the central term precisely enough, and we now distinguish three objects explicitly in Section 2. For each perturbation pair we define the relative SAE-code response $D_i$ (Eq. 3). The exact blind set is $S_0=\lbrace i \mid D_i=0\rbrace$ (Eq. 4); the margin set is $S_\gamma=\lbrace i \mid D_i<\gamma\rbrace$ (Eq. 5) with the coverage statistic $C_\gamma$ (Eq. 6); and $V_{\mathrm{Gini}}$ is the inequality of the finite response profile (Eq. 8). Table 1 states what each object measures and the role it plays. We make explicit that these are distinct. $S_0$ is typically empty for a continuous encoder (now stated as a proposition with proof), so $S_\gamma$ and the lower-tail magnitudes are the operational objects, while $V_{\mathrm{Gini}}$ describes profile shape rather than magnitude. The text now describes $V_{\mathrm{Gini}}$ as a measure of within-family response uniformity, not blind-spot severity, in line with the reviewer's reading.
>
> **2. Relation between the stabiliser condition and the metric.** We state $\operatorname{Stab}(G,F)\cap E=\lbrace e\rbrace$ (Eq. 2) as the exact, zero-response form of the blind-set principle, and we make the bridge to the metric explicit. The metric does not test this exact algebraic condition. Because the encoder is continuous, $D_i>0$ almost surely, so $S_0$ is empty with probability one and the exact stabiliser is already trivial; this is precisely why the operational analysis uses $S_\gamma$, the lower-tail magnitudes, and $V_{\mathrm{Gini}}$ rather than $S_0$. The blind-set condition defines the failure mode, the margin and lower-tail quantities define operational detectability, and the Gini statistic measures finite-profile imbalance. We have removed the use of $V<0.05$ and $V<0.005$ as detectability boundaries (see point 4).
>
> *(Continued in part 2.)*

---

> ### Author Response · Authors · 2026-07-04
> **Response part 3 of 4.**
>
> *(Point 7, continued.)*
>
> As a zero-cost between-family check using the existing absolute-response profiles, number swap lies at the low end of the Standard absolute-response distribution and below synonym in median response across all three models. $V$-regularisation raises the number-swap response in every model but does not reorder number swap above synonym, so the effect is best described as lifting a weak safety-relevant family rather than making it one of the highest-response families. This is consistent with the within-family design of $V_{\mathrm{Gini}}$ and with the absolute lower-tail metrics that carry the cross-family magnitude information.
>
> **8. Claim language and broader impact.** We reframed $V$-regularisation as a perturbation-sensitivity conditioning objective throughout. The blind-set condition defines the failure mode, the margin and lower-tail quantities define operational detectability, and the Gini term provides smooth training pressure on the finite response profile. The abstract no longer reports the earlier headline percentages, threshold counts are descriptive, and exact-zero values are reported with precision or dispersion. The Broader Impact statement now states explicitly that $V_{\mathrm{Gini}}$ measures within-family response uniformity rather than absolute task detectability, and that it should not be used as a standalone certification signal without absolute-sensitivity or downstream validation. The Discussion no longer suggests $V$ thresholds as a compliance or certification mechanism.
>
> **Strengthening 1. Load-bearing versus motivational role of the algebraic framework.** The reviewer is right that the method reduces operationally to a differentiable Gini term on perturbation responses, and that the manuscript should make clear how much of the algebraic apparatus this relies on. The minimal load-bearing element is the blind-set principle $\operatorname{Stab}(G,F)\cap E=\lbrace e\rbrace$ in its continuous, zero-response form (Eq. 2), which is what defines a blind spot and what the metric hierarchy in Table 1 operationalises. The single regularised basis used throughout the experiments needs nothing beyond this, and the self-contained chain from the blind-set condition, through the margin and lower-tail quantities that capture operational detectability, to the Gini statistic that provides differentiable training pressure requires no results beyond what is stated in the paper.
>
> The remaining algebraic material is motivational rather than load-bearing. The stabiliser monotonicity theorem appears only in the optional multi-factor extension (Appendix J), where it explains why adding parallel factors can only shrink the stabiliser and so motivates $m^\ast$ as a forward-looking design parameter for perturbation sets that a single basis cannot resolve. The reported result is $m^\ast=1$, so this extension is not exercised by any experiment in the paper. We have removed the broader $(H,V,\mathcal{G})$ design triad since the method does not rely on it; a single citation to the algebraic source is retained for provenance.
>
> **Strengthening 2. Consistency of the headline percentages.** We agree the earlier "83–100%" figure conflated different quantities. The abstract no longer reports it. Where percentage reductions appear, they are tied to a specific quantity and computation. The per-family imbalance reductions are reported against the metric they summarise (Table 8), and they are labelled as descriptive low-imbalance summaries rather than detectability figures. The training-loss trajectory and the per-family evaluation reductions are no longer reported under a single combined percentage.
>
> **Strengthening 3. Origin of the exact $V=0.000$ values.** Appendix P now explains why exact zeros arise. They reflect the interaction of the rounding convention with the $\varepsilon$ in Eq. 3 and with family size. For small or internally similar families the sorted response profile can become close to uniform, which drives the Gini statistic to values that round to $0.000$. We therefore report these cells either at full precision or as rounded near-zero values, and we now accompany them with seed dispersion (Table 2) and with the family-geometry controls of point 7, so that a near-zero $V_{\mathrm{Gini}}$ is not read as a trivial saturation effect.
>
> *(Continued in part 4.)*

---

> ### Author Response · Authors · 2026-07-04
> **Response part 4 of 4.**
>
> **Broader Impact.** We have acted on the specific concern about promoting $V$ as a compliance or certification signal. The Discussion no longer suggests that regulators specify $V$ thresholds or that developers demonstrate compliance through $V$ profiles. The Broader Impact statement now states explicitly that $V_{\mathrm{Gini}}$ measures within-family uniformity of sensitivity rather than absolute detectability, and that a system uniformly insensitive to a safety-relevant family could pass a $V$ threshold; accordingly, $V$ should not be used as a standalone certification signal without an absolute-sensitivity or downstream validation. The statement retains the cautions the reviewer found appropriate, namely that low $V$ on a tested family does not guarantee safety on untested families, and that the perturbation sets are synthetic and English-only. The Discussion now also states the underlying precondition explicitly, namely that $V$-regularisation reorients the feature basis but cannot create information the hidden state does not encode, so when a distinction is not represented at the chosen layer no change to the objective can recover it.
>
> **Title and figures.** Two presentation changes follow directly from the points above. First, the title is now "A symmetry-matching approach to blind-spot reduction in sparse autoencoders", replacing "elimination" with "reduction" to match a better claim language. Second, the figures have been replaced to reflect the corrected evidence. The submission's $V_{\mathrm{Gini}}$ comparison figure and the training-loss trajectory figure, which carried the earlier headline percentages, have been removed. In their place, Figure 1 shows the cumulative distribution of per-pair responses with the Standard low-response band marked, making the lower-tail lifting visible at the distributional level, and Figure 2 shows the frozen multi-seed separation of $V$-regularisation from the non-perturbation-aware objectives. The Gemma hyperparameter-sweep figure that accompanied the earlier floor analysis has been removed together with that analysis, as described above.
>
> We hope these revisions address the Reviewer's concerns. The definitional separation, the corrected extraction protocol, the absolute and multi-seed evidence, the independent probes, and the family-geometry controls together move the manuscript from the safety-framed blind-spot-elimination language of the submission to the narrower, evidence-supported claims the reviewer identified as well founded. The precision of the review has shaped this revision throughout, and we would warmly welcome the reviewer's further thoughts on the revised manuscript, together with any clarification should a point remain insufficiently addressed.

---

### Review · Reviewer_eLPT · 2026-06-30

**Summary Of Contributions:**

In this paper, the authors focus on the **timely and crucial issue of language models** becoming insensitive to critical changes, which can lead to fatal consequences in downstream tasks, e.g., clinical decision support. Based on the hypothesis that the *insensitivity* originates from the indistinguishability between features from 'original' (e.g., "the patient received 3 doses") and 'perturbed' (e.g., "the patient received 8 doses"), **the main contribution focuses on the measurement suggestion of the representational vulnerability, i.e., $V_{Gini}$.** The authors argued that $V_{Gini}$-based regularization significantly improves models' sensitivity against the fatal perturbations (e.g., a change from '3' to '8 doses'). Through extensive testing across models and perturbations, the metric $V$ is largely suppressed via adopting $V_{Gini}$ regularization.

**Audience:**

Yes

**Audience Explanation:**

**The problem should be highly emphasized in the literature**
- Notably, this work is not the first to observe the 'blind spot' of language models. At least, the authors have shown that existing approaches are probably not effective at addressing the pain point.
- To my understanding, the seemingly minor but crucial issue of language models should be highly emphasized to enhance the reliability of language models in sensitive application domains, e.g., medical treatments.
- Therefore, I believe that the topic would at least be of interest to some individuals in TMLR.

**Broader Impact Concerns:**

The broader impact has been sufficiently discussed in the '7. Conclusion. Broader Impact Statement' section.

**Claims And Evidence:**

No

**Claims Explanation:**

**1. Limited rigor of the suggested metric**
- First, the formulation of the metric, i.e., $V_{Gini}$ (eq. 2 in the main paper), is less rigorous. Technically, important ingrediants, including $G$, $F$, $E$, and $e$, should have been formally defined. Based on the understanding, $Stab(\cdot,\cdot)$ can be defined. Although it seems to be borrowed from (Balogh, 2026a), it should be recalled in the main text. Furthermore, $H$, $V$, and $\mathcal{G}$ are not fully formalized. In addition, $z_i$ seems to be a feature, but I cannot determine what it looks like, including whether it is a vector or a feature representation.
- Beyond the technical formulation flaws, more critically, I cannot catch any rationales why we should formulate vulnerability as shown in Eq 2, i.e., a reason for the "term1 minus term2" form; a reason for adopting $2\sum is_{(i)}/K\sum s_{(i)}$ term. This limited rigor makes the metric's design, which assesses the clarity of the main contribution, unconvincing.

**2. Relationship between 'performance' and 'metric'**
- Based on the metric design, a regularization term is proposed to directly suppress the metric. Then, the empirical testing is mainly exhibiting the 'suppressed metric, $V$'.
- It is quite trivial that the resulting metric values should be suppressed when we adopt it as a regularization term. More importantly, this behavior occurs regardless of the term's correctness.
- Thus, the important part is to prove the relationship between the suppressed metric and the improved performance against the perturbations. For example, if a model suppressed $V$, then is it true that the model can distinguish the numeric changes of 'doses' in the example case?

**Requested Changes:**

- Please revise the formulation to be more rigorous
- Please add the detailed reasons for the suggested metric design (referring to Reason of 'No' 1)
- Please show the numerical results of the clear relationship between the suggested metric and the resulting performance (e.g., robustness of models against minor yet cricial perturbations/changes)

---

> ### Author Response · Authors · 2026-07-02
> **Preliminary Response**
>
> We thank the reviewer for the careful and thorough reading of the manuscript.
>
> The review engages closely with the formulation of the metric and with the relation between the metric and downstream performance, and it also underlines that this class of language-model failure matters for sensitive application domains such as medical treatment. These comments have helped us see clearly where the manuscript needs to be strengthened, and we would like to outline the planned changes before submitting the revised manuscript and full response.
>
> **1. Formal definition of the metric and its ingredients**
>
> The reviewer is right that the submitted formulation did not define its ingredients precisely enough. The revision will add a self-contained treatment in Section 2. We plan to define the group of input transformations, the encoding, the task-relevant perturbation family, and the SAE code variables directly, together with an explicit set-builder statement of the stabiliser condition. The symmetry-matching condition would be recalled with attribution to its algebraic source in Balogh (2026a), so that the reader can follow the definitions without consulting external manuscripts. We also intend to make explicit that the experimental family is a finite sampled subset of the transformation family, which is the object all the finite-sample quantities are computed on.
>
> **2. Rationale for the metric design**
>
> The reviewer asks why the metric takes its specific form, and we accept that the submitted text did not make the reasons clear. The revision will introduce the metric first in its pairwise-difference form,
>
> $$
> V_{\mathrm{Gini}}(D)=\frac{1}{2K^2\bar D}\sum_{i=1}^{K}\sum_{j=1}^{K}\lvert D_i-D_j\rvert,
> $$
>
> the mean absolute pairwise difference of perturbation responses normalised by the mean response $\bar D$. In this form the metric states directly that it measures the inequality of the response profile across the family. We would then give the equivalent sorted closed form and derive it from the pairwise form through the rank multiplicity $2i-(K+1)$ of each ordered response, which should show that the rank weighting and the subtraction are consequences of the pairwise definition rather than design choices. The revised Section 2 will include this derivation in full.
>
> **3. Relation between the metric and perturbation performance**
>
> This is the reviewer's central point, that suppressing a regularised quantity is trivial unless it corresponds to better handling of the perturbations, and the dose example makes it concrete. It is a good question, and the submitted manuscript did not draw the distinction it calls for. The dose value is a token the model reads off the surface, so it is already separated and there is little for the objective to change there. The blind spot the paper is about appears on task-relevant directions that are not carried by such a surface token, where the weak-response lower tail is low, and that is where the method is meant to act. The revision will show this by evaluating endpoints that are not the training target, including the absolute lower-tail response, a null-calibrated target-versus-nuisance test, and held-out probes, and by reporting where the effect appears and where it does not. All in all, we appreciate the reviewer bringing this gap between metric and performance to light, since clarifying it strengthens the paper.
>
> **4. Claim language**
>
> Following some common concerns of all three reviews, we will reframe the method as a perturbation-sensitivity conditioning objective rather than a blind-spot eliminator. We would stress that this is a change of wording rather than a retreat from the underlying idea. The blind-set condition still defines the failure mode and does the same work in the forthcoming revised text, approximate margins define operational detectability, and the Gini term provides smooth training pressure on the finite response profile. What we withdraw is the stronger word "elimination", not the algebraic condition that motivates the method. As this ongoing work develops, we now consider a conditioning reading more accurate than the original elimination framing, and the revised title and abstract would reflect this change.
>
> These valuable comments of the reviewer have improved the rigour of the presentation, and in turn a point-by-point response with the corresponding definitions, tables, and uncertainty estimates will accompany the revised manuscript shortly. We hope that these planned changes will reduce the gap between the submitted version and what the review asks of it.
>
> Thank you very much for your time and effort in reviewing our paper.

---

> ### Author Response · Authors · 2026-07-04
> **Response part 1 of 2.**
>
> We thank the reviewer again for the careful reading and for pressing on the two points that most needed strengthening, the rigour of the formulation and the relation between the suppressed metric and actual perturbation performance. The revised manuscript is now uploaded, and we respond to each request in the order it was raised, indicating where each promised change landed.
>
> **What changed and what is preserved.** Because the revision is substantial, we state its scope up front. The original experiments and results all remain in the manuscript; nothing has been replaced, only corrected where an extraction-protocol audit required it. The claims have been narrowed rather than redirected. The title change from "elimination" to "reduction", announced in our earlier comment, reflects the narrowing that all three reviews converged on. Every added analysis answers a named reviewer request.
>
> **1. Rigour of the formulation.** Section 2 is now a self-contained formal treatment. We define the group $G$ of input transformations acting on the input space, the encoding $F$, the task-relevant perturbation family, and the SAE code variables directly in the paper, together with an explicit set-builder statement of the stabiliser condition $\operatorname{Stab}(G,F)=\lbrace g\in G \mid F(g\cdot x)=F(x)\ \text{for all}\ x\rbrace$ and the blind-set principle $\operatorname{Stab}(G,F)\cap E=\lbrace e\rbrace$ (Eqs. 1–2). The code object is stated explicitly. $z_i$ is the SAE code vector of the hidden state, and the per-pair response $D_i$ (Eq. 3) is the relative displacement of that vector under the perturbation. We also make explicit that the experimental family is a finite sampled subset of the transformation family, which is the object on which all finite-sample quantities are computed. The symmetry-matching condition is recalled with attribution to its algebraic source, but every definition needed to follow the paper is now given in the paper itself. A new proposition with proof shows that the exact blind set is empty almost surely for a continuous encoder, which is what motivates the margin set and the profile-imbalance level as the operational objects (Table 1 states the hierarchy).
>
> **2. Rationale for the metric design.** The revision introduces the metric first in its pairwise-difference form (Eq. 7),
>
> $$
> V_{\mathrm{Gini}}(D)=\frac{1}{2K^{2}\bar{D}}\sum_{i=1}^{K}\sum_{j=1}^{K}\lvert D_i-D_j\rvert,
> $$
>
> the mean absolute pairwise difference of perturbation responses normalised by the mean response $\bar{D}$. In this form the metric states directly what it measures, the inequality of the response profile across the family. Section 2 then derives the sorted closed form used in training (Eq. 8) from the pairwise form through the rank multiplicity $2i-(K+1)$ of each ordered response. This derivation shows that the "term1 minus term2" structure and the rank weighting are consequences of the pairwise definition, not design choices. The scale invariance $V_{\mathrm{Gini}}(cD)=V_{\mathrm{Gini}}(D)$ is stated formally (Eq. 9), together with its direct consequence, that the metric carries no information about absolute response magnitude, which is why the evaluation adds scale-sensitive endpoints.
>
> **3. Numerical relationship between the metric and perturbation performance.** This was the reviewer's central point, and the dose example makes it concrete, so we answer it in two layers, both now in the manuscript with numbers.
>
> *The dose value itself.* The digit sits on the surface of the text, so any downstream readout can recover it directly; there is little for a representation-level objective to change. We verified this rather than asserting it. A held-out dose-value probe (Appendix F) shows the Standard code already at ceiling, and $V$-regularisation does not improve it (the deltas are negative, $-0.04$ to $-0.16$ across models). The introduction now presents the dose example in this two-layer reading, with the surface token dependable at readout, while the weak lower tail concerns distinctions that no single surface token carries, such as negation, severity, or laterality.
>
> *(Continued in part 2.)*

---

> ### Author Response · Authors · 2026-07-04
> **Response part 2 of 2.**
>
> *(Point 3, continued.)*
>
> *Where the metric-performance link is tested.* The revision evaluates endpoints that are not the training target. (i) Absolute lower-tail response. $\Delta L_{20}(D)$ and $\Delta L_{20}(\lVert\Delta z\rVert)$ are positive across all three models under a frozen multi-seed protocol (Tables 2–3); by scale invariance the objective cannot by construction raise these magnitudes, so this is not a restatement of the trained quantity. (ii) A null-calibrated target-versus-nuisance detectability test (Table 5). The family-macro AUROC difference against meaning-preserving nuisance controls is positive on Gemma 2 and Qwen 2.5 with fixed-family intervals excluding zero, and uncertain in point direction on GPT-2, which we report as such. (iii) Held-out probes (Section 5.4). Point estimates improve on all three models, with template-bootstrap intervals that include zero, reported as suggestive rather than uniform. (iv) A zero-shot real-clinical audit on OpenI radiology minimal pairs (Table 9, Appendix B). Frozen general checkpoints, never trained on this data, show the lower-tail lift in all eight model-by-family cells (GPT-2 and Qwen 2.5, four clinical families each) with report-clustered intervals excluding zero; Gemma 2 enters the external evaluation through the laterality probe and the held-out family audit. (v) A dedicated paired experiment and a controlled non-saturated synthetic experiment (Appendices C–D) isolate the mechanism with common-randomness training, including the finding that on a task the Standard code already saturates, the focused objective does not help, the same boundary as the dose probe.
>
> The summary, stated in the text, is that $V$-regularisation reliably improves the geometry of weak response directions, while transfer to downstream probability margins is task- and model-dependent. It appears where there is representational headroom and not where a surface token already carries the distinction.
>
> **4. Claim language.** As announced, the method is reframed throughout as a perturbation-sensitivity conditioning objective rather than a blind-spot eliminator, and the title now reads "blind-spot reduction". This is a change of wording rather than a retreat from the underlying idea. The blind-set condition still defines the failure mode, approximate margins define operational detectability, and the Gini term provides smooth training pressure on the finite response profile. The abstract and contributions reflect the narrower claims, and the boundary cases above are reported alongside the positive results.
>
> We are grateful for the review's insistence on the metric-performance distinction; drawing it explicitly, and instrumenting it with endpoints outside the training objective, has made the paper's claims both narrower and better supported. We hope the revision addresses the concerns, and we would appreciate any further clarification if we have missed a point.

---

### Review · Reviewer_eFS4 · 2026-07-01

**Summary Of Contributions:**

This paper introduces **V-Gini**, a differentiable metric intended to quantify representational blind spots in sparse autoencoders (SAEs) by measuring the inequality of feature-space sensitivities across a perturbation family. Motivated by a symmetry-matching framework for perturbation awareness, the authors augment the standard SAE objective with a V-Gini regularization term that encourages more uniform responses to task-relevant perturbations.

The authors evaluate the approach on SAEs trained over hidden representations from GPT-2, Gemma 2, and Qwen 2.5. Across multiple perturbation families, the proposed regularizer substantially reduces the V-Gini metric relative to standard SAE training objectives, including L1, JumpReLU, and MDL-based variants, while requiring neither additional model capacity nor retraining of the underlying language model. The paper further investigates joint training across multiple perturbation families and analyzes the behavior of the method across architectures and hyperparameter settings.

**Additional Comments:**

This paper to me easily clears any bars to novelty and importance to the community; but is gated by the robustness and external validation being lacking. The number of models and datasets I do not have significant concern with, but the circularity and lack of any validation of the metric on any real world uses of SAE's concerns me as needing to be addressed. I believe these fixes are not ambiguous in nature though, and the authors may be able to do so in a reasonable time period.

**Audience:**

Yes

**Audience Explanation:**

I endorse that members of the TMLR audience would be interested in this work because it addresses a plausible and underexplored limitation of sparse autoencoders. Since SAEs are trained to faithfully reconstruct model activations, it is reasonable to ask whether representational blind spots in the underlying model are inherited by the learned feature space, and whether these blind spots can be mitigated through training.

That said, my interest is currently driven more by the problem formulation and proposed metric than by the empirical findings themselves. The paper convincingly demonstrates that the proposed objective can reduce the introduced $V_{\mathrm{Gini}}$ metric, but currently provides little extrinsic evidence that these reductions translate into meaningful improvements in perturbation awareness, interpretability, or downstream behavior. As a result, while I do believe the research direction is of interest, it is harder to assess the interest of the community in the findings till these gaps are validated.

**Broader Impact Concerns:**

None.

**Claims And Evidence:**

No

**Claims Explanation:**

I personally think $V_{\mathrm{Gini}}$ is a novel and compelling idea. Since SAEs are trained primarily to reconstruct activations, it is plausible that they inherit representational blind spots present in the underlying model, and explicitly encouraging perturbation sensitivity seems like a promising direction.

My main concern is that the empirical validation is largely circular. The paper introduces both the underlying sensitivity measure and the derived $V_{\mathrm{Gini}}$ metric, and nearly all of the experimental evaluation focuses on demonstrating improvements in these quantities under the proposed training objective. I tabulated the metrics reported throughout the paper:

| Item | Metric Tracked |
|--------|---------|
| Table 1 | Raw $V_{\mathrm{Gini}}$ |
| Figure 1 | Raw $V_{\mathrm{Gini}}$ |
| Table 2 | Raw $V_{\mathrm{Gini}}$ (plus reconstruction MSE) |
| Table 3 | Mean sensitivity |
| Figure 2 | Training-time $V_{\mathrm{Gini}}$ loss |
| Table 4 | Mean $V_{\mathrm{Gini}}$ across 16 families |
| Table 5 | Per-family $V_{\mathrm{Gini}}$ |
| Figure 3 | Minimum $V_{\mathrm{Gini}}$ reached during training |

While these results convincingly demonstrate that the proposed objective successfully optimizes the proposed metric, they provide no extrinsic evidence that reductions in $V_{\mathrm{Gini}}$ correspond to meaningful improvements in perturbation awareness, interpretability, or downstream behavior. The connection is intuitively plausible, but I would have liked to see at least a few external validation metrics or experiments that are not directly derived from the proposed framework.

A second concern, primarily about presentation rather than correctness, is the manuscript's reliance on Balogh (2026a,b). The symmetry-matching framework plays a central role in motivating the method and interpreting the results, yet many of its key concepts are imported by reference. Since this is not a standard framework familiar to most readers of the SAE or interpretability literature, I believe the paper would benefit from a brief self-contained section (or appendix) summarizing the relevant definitions, intuitions, and theoretical claims needed to understand the present work without consulting external manuscripts.

A related issue is that the paper contains relatively little qualitative analysis of the blind spots it seeks to measure and eliminate. For example, it would be useful to inspect concrete SAE features that exhibit high versus low $V_{\mathrm{Gini}}$, identify cases where semantically distinct concepts or symbols are confounded in feature space, and show how the proposed regularizer changes these representations. Such examples would help build intuition for what the metric is capturing, why certain perturbation families induce blind spots, and whether reductions in $V_{\mathrm{Gini}}$ correspond to qualitatively meaningful changes in the learned feature representations.

**Requested Changes:**

I segment these by **critical** and **would strengthen**:

### Critical

1. **Provide extrinsic validation of the proposed metric.**

   The current evaluation convincingly demonstrates that the proposed objective reduces $V_{\mathrm{Gini}}$ and related sensitivity-based quantities. However, essentially all reported metrics are either $V_{\mathrm{Gini}}$ itself or quantities directly derived from the same underlying sensitivity framework. As a result, it remains unclear whether improvements in $V_{\mathrm{Gini}}$ correspond to meaningful improvements in the behavior or usefulness of the learned SAE representations.

   More generally, I would like to see evidence that $V_{\mathrm{Gini}}$ is not merely a quantity that can be optimized, but one whose optimization produces representations with desirable properties outside the framework used to define it.

   In particular, recent efforts such as **SAEBench** emphasize evaluating SAEs across a broader collection of criteria, while systems such as **Delphi** have enabled increasingly systematic analyses of feature quality and interpretability. At the same time, one possible contribution of this work is precisely that existing evaluation frameworks may not adequately capture the class of blind spots targeted by $V_{\mathrm{Gini}}$. Consequently, I do not view evaluation against existing SAE benchmarks as strictly necessary, although it would be a nice and standard measurement to make for an SAE improvement paper.

   Rather, I would like to see some form of independent validation that the blind spots identified by the proposed framework correspond to meaningful representational deficiencies, and that reducing $V_{\mathrm{Gini}}$ measurably mitigates those deficiencies. Existing SAE evaluation frameworks provide one possible route to such evidence, but they are by no means the only one.

   I do not have a strong preference for any particular validation strategy, nor do I intend the list below as a checklist of required experiments. Rather, I am offering a range of possible directions that could provide independent evidence for the practical significance of $V_{\mathrm{Gini}}$. I would be satisfied by any subset of these that provides a convincing validation approach, including alternatives not listed here.

   Examples could include:

   a. **Human annotation studies.**
   Use human annotators to evaluate whether lower-$V_{\mathrm{Gini}}$ SAEs produce features that are more semantically coherent, easier to interpret, or better separated with respect to the targeted perturbation families. Since the central claim is that certain representational blind spots are meaningful and undesirable, human judgment provides a particularly strong source of independent validation.

   b. **Interpretability and feature-quality evaluation.**
   Evaluate whether lower-$V_{\mathrm{Gini}}$ SAEs produce features that are easier to interpret, less polysemantic, more semantically coherent, or otherwise exhibit improved feature quality. This could be assessed through auto-interpretability pipelines, Delphi-style analyses, or related methodologies.

   c. **Targeted blind-spot case studies.**
   Identify specific perturbation families, symbols, concepts, or representation-space confusions that motivate the framework, and demonstrate that high-$V_{\mathrm{Gini}}$ cases correspond to genuinely confounded representations while low-$V_{\mathrm{Gini}}$ cases are better separated. Such analyses would help establish that the metric tracks meaningful representational phenomena rather than only a formal sensitivity statistic.

   d. **Classifier or probing experiments.**
   If the motivating claim is that certain perturbations become more distinguishable, evaluate whether simple probes or classifiers can more accurately distinguish those perturbations from SAE representations after regularization.

   e. **Feature intervention studies.**
   Demonstrate that features become more targeted, disentangled, or causally meaningful through feature ablations, steering experiments, activation patching, or related intervention-based analyses.

   f. **Evaluation against existing SAE criteria.**
   Where applicable, investigate whether reductions in $V_{\mathrm{Gini}}$ correlate with established SAE quality measures (e.g., interpretability, concept detection, feature disentanglement, or downstream utility), while recognizing that the proposed blind spots may not be fully captured by existing benchmarks.

   g. **Correlation studies.**
   Demonstrate that $V_{\mathrm{Gini}}$ correlates with independently meaningful quantities, providing evidence that optimizing the metric corresponds to improvements beyond the metric itself.

2. **Include qualitative case studies of blind spots and their mitigation.**

   The paper would benefit from concrete examples showing what a blind spot looks like in practice. For example, examining individual SAE features before and after regularization, demonstrating confounded concepts or symbolic perturbations, and illustrating how the proposed objective changes the learned representation would provide useful intuition for what $V_{\mathrm{Gini}}$ is measuring.

### Would Strengthen
3. **Clarify the scope of the claims if such validation is not provided.**

   If additional experiments are not feasible, the manuscript should more carefully distinguish between demonstrating improvement in the proposed metric and demonstrating broader improvements in representation quality, perturbation awareness, reliability, or interpretability.


4. **Provide a more self-contained summary of the symmetry-matching framework.**

   The manuscript relies heavily on Balogh (2026a,b). Since these ideas are central to the motivation and interpretation of the results but are not yet standard within the SAE or interpretability literature, a concise overview of the relevant definitions, intuitions, and theoretical claims would improve accessibility and reduce the need for readers to consult external manuscripts.

5. **Discuss limitations and applicability of the perturbation families considered.**

   A brief discussion of how sensitive the conclusions are to the choice of perturbation family, and whether the authors expect the approach to generalize to broader classes of representational blind spots, would help contextualize the results.


### Minor comments:
1.  Giving the pairwise difference definition of $V_{\mathrm{Gini}}$ early (although equivalent) would give the intuition more immediately to reviewers who haven’t come across gini coefficient before.

---

> ### Author Response · Authors · 2026-07-02
> **Preliminary Response**
>
> We are grateful for the thoughtful and encouraging review, and for the recognition of the interest of the problem and the proposed metric. The central concern about circularity is well taken, and it has shaped the direction of the revision. We summarise the planned changes here, ahead of the revised manuscript and full response.
>
> **1. Extrinsic validation beyond the defining framework**
>
> The reviewer puts the main concern well, that most of the reported metrics are either the trained quantity or quantities derived from the same sensitivity framework, so a reduction of the metric is not on its own evidence of improvement. The revision will add endpoints measured outside the training objective. These would include a null-calibrated target-versus-nuisance detectability test, held-out probes, a real-clinical lower-tail response on out-of-distribution radiology minimal pairs evaluated zero-shot on frozen checkpoints, and a controlled paired experiment in a non-saturated regime. We also intend to make explicit that the scale-invariant objective cannot by construction raise the absolute response magnitude. In preliminary analyses on the real-clinical data the absolute lower-tail response and the trained inequality statistic already appear to move independently, which would address the circularity concern directly. We would also be candid that these gains are model- and task-dependent. On tasks where the standard code already separates the distinction, there is nothing for the objective to recover and it does not help, and we plan to report these cases alongside the positive ones. The revised manuscript will present the full set of analyses with uncertainty estimates.
>
> **2. Qualitative case study of blind spots**
>
> The reviewer would like to see what a blind spot looks like in practice, and we think a concrete, qualitative view would indeed help build intuition for what the metric captures. The revision will add a case study on real radiology minimal pairs that shows what a weak response looks like under the Standard SAE and how the lower-tail response changes under the conditioning objective. We plan to keep this analysis at the level of the response profile. Since we do not include a feature-activation study, we would not claim that individual SAE units become monosemantic, and we intend to state this scope explicitly.
>
> **3. Self-contained summary of the symmetry-matching framework**
>
> The reviewer notes that the symmetry-matching framework is not yet standard in the SAE or interpretability literature, and we accept that it should be readable without consulting external manuscripts. The revision will add a self-contained section that defines the group of input transformations, the encoding, the perturbation family, the SAE code variables, and an explicit set-builder statement of the stabiliser condition, while keeping the algebraic motivation borrowed from Balogh (2026a). With the definitions given in the paper, the exposition is self-contained and no longer depends on external manuscripts. We also plan to introduce the metric in its pairwise-difference form early, as the reviewer suggested,
>
> $$
> V_{\mathrm{Gini}}(D)=\frac{1}{2K^2\bar D}\sum_{i=1}^{K}\sum_{j=1}^{K}\lvert D_i-D_j\rvert,
> $$
>
> the mean absolute pairwise difference of perturbation responses normalised by the mean response $\bar D$, which gives the inequality intuition immediately. The equivalence of this form with the sorted expression used in training, which we outline in our response to Reviewer eLPT, will appear in the revised Section 2.
>
> **4. Scope of the claims and the perturbation families**
>
> Following some common concerns of all three reviews, we will reframe the method as a perturbation-sensitivity conditioning objective rather than a blind-spot eliminator, and we plan to add a discussion of how sensitive the conclusions are to the choice of perturbation family. As this line of work develops, the results so far lead us to think that a conditioning reading fits the evidence better than the original elimination framing. We would note that this reframing softens the claim language without discarding the symmetry-matching condition, which still carries the definition of the failure mode in the revised text. We intend to report the boundary cases directly, including tasks where the Standard code is already saturated and a focused objective does not help, so that the scope of the positive results is clear.
>
> The clarity of the review made the required changes easy to identify. We will upload the revised manuscript together with a detailed response addressing each point in turn. Thank you very much for your time and effort in reviewing our paper. Should we have missed any part of your comments, we would highly appreciate your kind clarification.

---

> > ### Comment · Reviewer_eFS4 · 2026-07-04
> > **I look forward to the promised revisions.**
> >
> > In particular, the items (1) and (2) - a careful qualitative study and extrinsic validation - will both strengthen my confidence in the results.

---

> ### Author Response · Authors · 2026-07-04
> **Response part 1 of 2.**
>
> We thank the reviewer again for the constructive review and for the follow-up note confirming that extrinsic validation and a careful qualitative study are the two items that would most strengthen confidence in the results. The revised manuscript is now uploaded and those two items are the core of the revision, and we respond point by point, indicating where each promised change takes effect and what the numbers are.
>
> **What changed and what is preserved.** Because the revision is substantial, we state its scope up front. The original experiments and results all remain in the manuscript, nothing has been replaced, only corrected where an extraction-protocol audit required it. The claims have been narrowed rather than redirected, from blind-spot elimination to perturbation-sensitivity conditioning, and the title reflects this. Every added analysis answers a named reviewer request an the new supporting material is placed in appendices, and the boundary cases are reported alongside the positive results.
>
> **Critical 1. Extrinsic validation of the proposed metric.** The revision adds endpoints measured outside the training objective, following several of the directions the reviewer listed (case studies, probing, correlation-style independence evidence), and reports where the effect appears and where it does not.
>
> *Structural independence.* We first make explicit why the added endpoints are not derivable from the trained quantity. $V_{\mathrm{Gini}}$ is scale-invariant (Eq. 9), so it carries no information about absolute response magnitude, and the objective cannot by construction raise the absolute lower tail. Any absolute lift is therefore an extrinsic outcome, not a restatement of the optimised metric.
>
> *Real-clinical zero-shot audit (Table 9, Appendix B).* Frozen general checkpoints, trained only on synthetic template families and OpenWebText, are evaluated zero-shot on out-of-distribution OpenI radiology minimal pairs (negation, severity, laterality, anatomical direction). The lower-tail response $L_{20}(D)$ rises in all eight model-by-family cells (GPT-2 and Qwen 2.5, four clinical families each), with report-clustered 95% intervals of $\Delta L_{20}$ excluding zero. On this data the lift and the trained inequality statistic move independently, which addresses the circularity concern directly. The audit also reports the accompanying perturbation-domain reconstruction cost on Qwen 2.5 (general OWT reconstruction is unchanged), so the trade-off appears alongside the gain.
>
> *Null-calibrated detectability and probes (reviewer's direction d; Section 5.4).* A target-versus-nuisance separation test against meaning-preserving nuisance controls is positive on Gemma 2 and Qwen 2.5 with fixed-family intervals excluding zero, and uncertain in point direction on GPT-2. Held-out negation-template probes improve in point estimate on all three models with intervals that include zero, and an external OpenI laterality probe shows small deltas that cross zero on GPT-2 and Qwen 2.5, while the Gemma 2 delta is negative with an interval excluding zero on this saturated external task. These enter the paper as validation evidence with their uncertainty, not as a safety guarantee.
>
> *Controlled isolation (Appendices C–D).* A dedicated paired experiment with common randomness (shared initialisation and minibatch order) and a controlled non-saturated synthetic experiment isolate the mechanism. Where headroom exists the conditioning objective lifts the weak tail and improves discriminability. Where the Standard code is already saturated it does not help therefore those cases appear in the scope paragraph and in the appendices below.
>
> *Negative and null results, reported directly.* The dose-value probe is at ceiling and $V$-reg does not improve it (Appendix F, negative deltas) as a natural-negation classification test on properly powered real data is null (scope paragraph of Section 6). Saturation preflights are reported (Appendix H). The scope paragraph states that gains are model- and task-dependent.
>
> On SAEBench-style benchmark evaluation, as the reviewer anticipated, we did not make it the centrepiece, because the blind spots targeted here are family-specific response deficiencies that generic benchmarks do not directly instrument. We consider the zero-shot real-data audit and the null-calibrated tests the more direct form of independent evidence for this claim, and we agree broader benchmark evaluation is a natural next step.
>
> *(Continued in part 2.)*

---

> ### Author Response · Authors · 2026-07-04
> **Response part 2 of 2.**
>
> **Critical 2. Qualitative case study of blind spots.** Appendix B contains the case study on real radiology minimal pairs. It shows concrete report pairs on which the Standard SAE's code response sits in the weak lower tail (the perturbation flips the clinical meaning while the code barely moves) and how the response profile changes under the conditioning objective on the same frozen pairs. As announced, the analysis is kept at the level of the response profile. We do not include a feature-activation study, we do not claim that individual SAE units become monosemantic, and this scope is stated explicitly in the appendix.
>
> **Would strengthen 1. Scope of the claims.** Addressed throughout. The method is presented as a perturbation-sensitivity conditioning objective, the title now reads "blind-spot reduction", and the distinction between improving the proposed metric and demonstrating broader representation-quality improvements is drawn explicitly in Section 5.4 and the Discussion, with the negative cases listed above reported in the main text.
>
> **Would strengthen 2. Self-contained summary of the symmetry-matching framework.** Section 2 now defines the group of input transformations, the encoding, the perturbation family, and the SAE code variables directly, with a set-builder statement of the stabiliser condition and an explicit statement of which parts of the algebraic framework are load-bearing (the blind-set condition and the metric hierarchy of Table 1) versus motivational (the multi-factor material, now confined to Appendix J). The paper is readable without consulting the external manuscripts while a single citation to the algebraic source is retained for provenance.
>
> **Would strengthen 3. Limitations and applicability of the perturbation families.** Beyond the discussion paragraph, we turned this into an experiment. A held-out family generalisation audit (Appendix E) evaluates the frozen checkpoints zero-shot on a pool of one hundred perturbation families. The pool metadata records a creation date of 25 May 2026, before the first review was posted, and the held-out status does not rest on that timestamp. After a disclosed duplicate audit that counts set-equal shard families once, seventy-two distinct held-out families remain in two pre-specified tiers, namely unseen instances and lexicons of trained family kinds (54 families), and family kinds structurally unlike anything in training, covering operator-level, inflectional, syntactic, and discourse-level transformations (18 families). The lower-tail lift reproduces on both tiers on all three models (CI-positive fractions 91%/100% on GPT-2, 89%/89% on Qwen 2.5, 100%/100% on Gemma 2), no family is significantly negative on any model, and the exceptions concentrate exactly where a surface token carries the change (numeric shards on Qwen), matching the dose-value ceiling. The conclusions therefore do not hinge on the choice of the sixteen trained families, and the approach extends to broader classes of representational blind spots than those used in training.
>
> **Minor comment.** The metric is now introduced first in its pairwise-difference form (Eq. 7), with the sorted training form derived from it through the rank multiplicity of each ordered response, so the inequality intuition comes before the closed form.
>
> We are grateful for the direction the review gave the revision; the circularity concern in particular is what led to the zero-shot real-data audit and the held-out family audit, which we believe are now among the strongest results in the paper. We hope the revision addresses the concerns, and we would appreciate any further clarification if we have missed a point.